# The *Arabidopsis* Cdk1/Cdk2 homolog CDKA;1 controls chromosome axis assembly during plant meiosis

Chao Yang[1], Kostika Sofroni[1] (iD), Erik Wijnker[1,†], Yuki Hamamura[1], Lena Carstens[1,‡],
Hirofumi Harashima[2,§], Sara Christina Stolze[3], Daniel Vezon[4], Liudmila Chelysheva[4] (iD),
Zsuzsanna Orban-Nemeth[5,¶], Gaëtan Pochon[1], Hirofumi Nakagami[3] (iD), Peter Schlögelhofer[5],
Mathilde Grelon[4] & Arp Schnittger[1,*] (iD)

## Abstract

Meiosis is key to sexual reproduction and genetic diversity. Here, we show that the *Arabidopsis* cyclin-dependent kinase Cdk1/Cdk2 homolog CDKA;1 is an important regulator of meiosis needed for several aspects of meiosis such as chromosome synapsis. We identify the chromosome axis protein ASYNAPTIC 1 (ASY1), the *Arabidopsis* homolog of Hop1 (homolog pairing 1), essential for synaptonemal complex formation, as a target of CDKA;1. The phosphorylation of ASY1 is required for its recruitment to the chromosome axis via ASYNAPTIC 3 (ASY3), the *Arabidopsis* reductional division 1 (Red1) homolog, counteracting the disassembly activity of the AAA⁺ ATPase PACHYTENE CHECKPOINT 2 (PCH2). Furthermore, we have identified the closure motif in ASY1, typical for HORMA domain proteins, and provide evidence that the phosphorylation of ASY1 regulates the putative self-polymerization of ASY1 along the chromosome axis. Hence, the phosphorylation of ASY1 by CDKA;1 appears to be a two-pronged mechanism to initiate chromosome axis formation in meiosis.

**Keywords** CDKA;1; ASY1; ASY3; chromosome axis; PCH2
**Subject Categories** Cell Cycle; Plant Biology
**The EMBO Journal (2020) 39: e101625**

## Introduction

Cell division relies on a highly orchestrated order of events to allow the faithful distribution of chromosomes to daughter cells. Progression through the cell cycle is controlled by the activity of cyclin-dependent kinases (Cdks; Morgan, 1997; Malumbres *et al*, 2009; Harashima *et al*, 2013). Eukaryotes usually contain several different families of cyclins that are thought to provide substrate specificity to Cdk–cyclin complexes and guide their intracellular localization (Miller & Cross, 2001; Pagliuca *et al*, 2011). However, the absolute levels of kinase activity have been found to be of key importance for cell cycle control, and at least in fission yeast, a single Cdk–cyclin complex is sufficient to drive both mitosis and meiosis (Coudreuse & Nurse, 2010; Gutiérrez-Escribano & Nurse, 2015).

In comparison with mitosis, much less is known about how Cdks control the progression of the two consecutive division events of meiosis. Meiosis II leads to the separation of sister chromatids that, at least formally, resembles a mitotic division and is thought to largely rely on similar control mechanisms as mitosis. In contrast, meiosis I holds many features that are not known from mitosis, foremost recombination between homologous chromosomes. Nonetheless, Cdk–cyclin complexes have been shown to control several aspects of meiosis I such as the formation of DNA double-strand breaks (DSBs) at the beginning of the meiotic recombination process by phosphorylating Mer2/Rec107 (meiotic recombination 2/recombination 107; Rockmill & Roeder, 1990; Henderson *et al*, 2006; Li *et al*, 2006).

Furthermore, the repair of DSBs through meiotic recombination has been found to involve Cdks, namely to phosphorylate the nuclease Sae2/Com1 (sporulation in the absence of spo eleven 2/completion of

1 Department of Developmental Biology, University of Hamburg, Hamburg, Germany
2 RIKEN Center for Sustainable Resource Science, Yokohama, Japan
3 Max-Planck-Institute for Plant Breeding Research, Cologne, Germany
4 Institut Jean-Pierre Bourgin, INRA, AgroParisTech, CNRS, Université Paris-Saclay, Versailles, France
5 Department of Chromosome Biology, Max F. Perutz Laboratories, Vienna Biocenter, University of Vienna, Vienna, Austria
*Corresponding author. Tel: +49 40 428 16 502; Fax: +49 40 428 16 503; E-mail: arp.schnittger@uni-hamburg.de
†Present address: Laboratory of Genetics, Wageningen University & Research, Wageningen, The Netherlands
‡Present address: Plant Developmental Biology & Plant Physiology, Kiel University, Kiel, Germany
§Present address: Solution Research Laboratory, AS ONE Corporation, Kawasakiku, Kawasaki, Japan
¶Present address: Institute of Molecular Pathology, Vienna Biocenter, Vienna, Austria
[The copyright line of this article was changed on 16 December 2019 after original online publication.]

meiotic recombination 1) and by that promotes its activity to generate 3′ overhangs at the DSB site (Huertas & Jackson, 2009; Anand *et al*, 2016; Cannavo *et al*, 2018). These DNA ends are further processed by the MRN/MRX complex comprising the subunits Mre11 (meiotic recombination 11), Rad50 (radiation 50), and Nbs1/Xrs2 (Nijmegen breakage syndrome 1/X-ray sensitive 2) (Mimitou and Symington, 2009; Manfrini *et al*, 2010). Subsequently, the single DNA strands are bound by the recombinases Rad51 (radiation 51) and Dmc1 (disrupted meiotic cDNA1) to promote strand invasion and formation of heteroduplex DNA (Shinohara *et al*, 1997; Kurzbauer *et al*, 2012; Da Ines *et al*, 2013). Depending on how the subsequently resulting double Holliday junctions are resolved, meiotic crossovers (COs) can be formed that lead to the reciprocal exchange of DNA segments between homologous chromosomes (Zickler & Kleckner, 2015; Lambing *et al*, 2017). Cdks were found to partially co-localize with Rad51 as well as other components acting downstream of Rad51 involved in CO formation (Baker *et al*, 1996; Zhu *et al*, 2010). This, together with the observation that inhibition of Cdk activity in early meiosis abolished the formation of Rad51 foci, led to the conclusion that the activity of Cdk is essential for DSB formation and/or processing (Henderson *et al*, 2006; Huertas *et al*, 2008; Zhu *et al*, 2010).

In many species, the synaptonemal complex (SC) stabilizes the pairing of homologous chromosomes and plays an important role in promoting the interhomolog bias during recombination and in maturation of recombination intermediates into COs (Zickler & Kleckner, 1999; Mercier *et al*, 2015). The SC is formed by the two proteinaceous axes of homologous chromosomes that will become then the lateral elements of the SC after synapsis. A number of proteins have been identified that are required for the correct formation of the chromosome axis. These include Red1 in yeast and its orthologs such as ASY3 in *Arabidopsis* (Rockmill & Roeder, 1990; Smith & Roeder, 1997; Ferdous *et al*, 2012). Another key protein of the chromosome axis is the HORMA domain protein Hop1 in yeast and its ortholog ASY1 in *Arabidopsis* (Hollingsworth *et al*, 1990; Aravind & Koonin, 1998; Armstrong, 2002). The phosphorylation of Hop1 at an [S/T]Q cluster domain by Tel1 (Telomere maintenance 1) and Mec1 (mitosis entry checkpoint 1), the ATM (ataxia telangiectasia mutated) and ATR (ataxia telangiectasia and Rad3-related) orthologs, is essential for the interhomolog-biased recombination, but not for the chromosomal loading of Hop1 (Carballo *et al*, 2008).

For the correct assembly of the SC, Hop1/ASY1 is recruited to the axis by interaction with Red1/ASY3 (Bailis & Roeder, 1998; de los Santos & Hollingsworth, 1999; Ferdous *et al*, 2012). Furthermore, it was recently proposed that Hop1 might build a homopolymer through its C-terminal closure motif and it was thought that this polymerization is likely crucial for its function and axis association since the point mutation K593A in the closure motif of Hop1 causes an 11-fold reduction in CO number and results in high spore lethality (Niu *et al*, 2005; West *et al*, 2018).

In wild type, the chromosome axes (lateral elements) of homologs become connected in the SC via the central region formed by dimers of the Zip1/ZYP1 family of proteins along with other components (Zickler & Kleckner, 2015). SC assembly goes along with the coordinated release of Hop1/ASY1 from the chromosome axis, catalyzed by the triple AAA$^+$ ATPase PCH2 (Wojtasz *et al*, 2009; Chen *et al*, 2014; Lambing *et al*, 2015). However, it is not clear how the dynamic localization Hop1/ASY1 on chromosomes is regulated.

Cdk complexes have also been implicated in the assembly of the SC since mutations in their catalytic core, i.e., in *Cdk2* in mice and in *CDC28* (*Cdk1* homolog) in budding yeast, resulted in defects in SC formation (Ortega *et al*, 2003; Zhu *et al*, 2010). However, although Zip1 has been shown to be phosphorylated by Cdk complexes *in vitro*, the molecular details of Cdk function for SC formation are still obscure since the SC is assembled normally in *zip1* mutants in which the Cdk phosphorylation sites were exchanged with amino acids that cannot be phosphorylated (Zhu *et al*, 2010).

The model plant *Arabidopsis*, similar to other multicellular eukaryotes, has several Cdks and cyclins with some of them having been assigned a function in meiosis (Wijnker & Schnittger, 2013). Six out of the 10 A- and one out of the nine B-type cyclins are expressed in meiosis including SOLO DANCERS (SDS), an atypical cyclin that has similarities to both A- and B-type cyclins (Azumi *et al*, 2002; Bulankova *et al*, 2013). However, of these eight cyclins potentially involved in meiosis, only the loss of either *CYCA1;2*, also known as *TARDY ASYNCHRONOUS MEIOSIS* (*TAM*), *CYCB3;1*, or *SDS* was found to result in meiotic defects (Magnard *et al*, 2001; Azumi *et al*, 2002; d'Erfurth *et al*, 2010; Bulankova *et al*, 2013; Prusicki *et al*, 2019). TAM is required for the repression of meiotic exit after the first meiotic division and the timely progression through meiosis II. SDS is necessary for crossover (CO) formation after DSBs have been induced, and the meiotic recombinase DMC1 does not localize to chromosomes in *sds* mutants (De Muyt *et al*, 2009). Mutants in *CYCB3;1* have only a weak mutant phenotype and occasionally show premature and ectopic cell wall formation during meiosis I, a phenotype, however, that can be strongly enhanced in double mutants with *sds* demonstrating a redundant function of at least some of the meiotic cyclins in *Arabidopsis* (Bulankova *et al*, 2013).

SDS and TAM build active kinase complexes with CDKA;1, the *Arabidopsis* Cdk1/Cdk2 homolog, that is the main cell cycle regulator in *Arabidopsis* (Cromer *et al*, 2012; Harashima & Schnittger, 2012; Nowack *et al*, 2012; Cifuentes *et al*, 2016). A function of CDKA;1 in meiosis is supported by the analysis of weak loss-of-function mutants, which are completely sterile (Dissmeyer *et al*, 2007, 2009). Next to CDKA;1, CDKG has been implicated in meiosis by controlling synapsis at ambient but not low temperatures (Zheng *et al*, 2014). However, CDKG, which is related to human Cdk10, is likely involved in transcriptional and posttranscriptional control of gene expression and presumably does not control structural components of chromosomes directly (Doonan & Kitsios, 2009; Tank & Thaker, 2011; Huang *et al*, 2013; Zabicki *et al*, 2013).

Here, we demonstrate by detailed cytological and genetics studies that CDKA;1 is an important regulator of meiosis especially for chromosome synapsis and bivalent formation. We show that ASY1 is a phosphorylation target of CDKA;1 and that the phosphorylation of ASY1 is crucial for chromosomal axis formation in *Arabidopsis* by two, possibly interconnected mechanisms, involving the binding to ASY3 as well as to itself leading to ASY1 polymers assembling along the chromosome axis.

## Results

### Changes in subcellular distribution of CDKA;1 during meiosis

For a detailed understanding of the role of CDKA;1 in meiosis, we first analyzed its localization pattern in male meiocytes. Previous

studies using a functional fusion of CDKA;1 to mVenus have shown that CDKA;1 is present in both female meiosis and male meiosis (Nowack *et al*, 2007; Bulankova *et al*, 2010; Zhao *et al*, 2012). Since the previous reporter was subject to frequent silencing effects, a new *CDKA;1* reporter was generated not relying on the cDNA, as in the previous construct. Instead, a 7 kb genomic fragment into which mVenus was introduced before the stop codon of *CDKA;1* was used. The expression of this construct fully rescued the *cdka;1* mutant phenotype and gave rise to stable CDKA;1:mVenus expression (Fig EV1A–C).

By using this reporter, the subcellular localization pattern of CDKA;1 during male meiosis was revealed (Fig 1A and B, and Movie EV1). In early prophase, CDKA;1:mVenus is localized in both the nucleus (~60–70%) and the cytoplasm (~30–40%). As prophase progresses, CDKA;1 accumulates more strongly in the nucleus (~80%). Then, toward the end of prophase, CDKA;1 becomes more cytoplasmically localized (~50%). After nuclear envelope breakdown, CDKA;1 decorates the first meiotic spindle and later accumulates in the two forming nuclei. In metaphase II, CDKA;1 is uniformly present in the entire cell, then is enriched at the spindle, and subsequently accumulates in the nuclei of the four meiotic products, i.e., the microspores (Fig 1A).

Due to the strong accumulation in the nucleoplasm, the presence of CDKA;1 at chromosomes, as reported for its mouse homolog Cdk2 or its yeast homolog Cdc28 (Ashley *et al*, 2001; Zhu *et al*, 2010), was difficult to judge. To address the chromosomal localization pattern of CDKA;1, we used plants that express a *StrepIII-tag-CDKA;1* fusion construct known to completely rescue the *cdka;1* mutant phenotype (Pusch *et al*, 2012), and followed the CDKA;1 localization in meiosis by immunolocalization using ASY1, a key component of the chromosome axis, for staging of meiosis. While Cdk2 and Cdc28 show a distinct punctuate staining in meiosis in mice and yeast (Ashley *et al*, 2001; Zhu *et al*, 2010), our experiments revealed that CDKA;1 co-localizes with ASY1 and forms a continuous signal along chromosomes at leptotene. At zygotene, when homologous chromosomes start to synapse, the fluorescent signals for both reporters, ASY1 and CDKA;1, concomitantly disappeared from the chromosome axes (Fig 1C). Since ASY1 is specifically removed from the synapsed chromosomes, we conclude from the similar patterns of CDKA;1 that CDKA;1 is excluded from the synapsed regions. These data suggest that CDKA;1 physically interacts with the chromosome axis during early meiotic prophase and might be important for chromosome pairing and synapsis.

## Meiosis is severely affected in hypomorphic *cdka;1* mutants

To assess the requirement of CDKA;1 for early stages of meiosis, we compared meiotic progression by chromosome spreads between wild-type plants and two previously described weak loss-of-function *cdka;1* mutants (Figs 1D and EV1D). These alleles resulted from the complementation of a *cdka;1* null mutant with *CDKA;1* expression constructs, in which conserved amino acids have been replaced resulting in CDKA;1 variants with strongly reduced kinase activity: *cdka;1 PRO_{CDKA;1}:CDKA;1^{T161D}* (in the following designated *CDKA;1^{T161D}*) and *cdka;1 PRO_{CDKA;1}: CDKA;1^{T14D;Y15E}* (in the following referred to as *CDKA;1^{T14D;Y15E}* (Dissmeyer *et al*, 2007, 2009). Both mutants were found to exhibit

similar meiotic phenotypes during male meiosis, because of which we focus on the description of one allele (*CDKA;1^{T161D}*) in the following (Figs 1D and EV1D).

In wild-type meiosis, chromosomes start to condense during early prophase, and initiate chromosome synapsis during zygotene, leading to full homolog synapsis at pachytene. Chromosome morphology becomes diffuse at diplotene followed by chromosome re-condensation toward diakinesis when bivalents become visible (Fig 1D a–c).

In *CDKA;1^{T161D}*, the first difference from the wild type becomes notable at zygotene-like stage manifested by the presence of clear thread-like chromosomes and the accumulation of mitochondria at the side of the meiocytes in which no homolog synapsis is observed (Fig 1D h) (58%; *n* = 120). The absence of synapsis was confirmed by the failure of ZYP1, a component of the central region of the synaptonemal complex, to localize to chromosomes of male meiocytes of *CDKA;1^{T161D}* mutants as revealed by immunofluorescence analysis (Fig EV1E). Pachytene-like stages of *CDKA;1^{T161D}* meiocytes show the characteristically even distribution of mitochondria as that in wild type through the cell, but have largely unpaired chromosomes (Fig 1D i). Like in the wild type, chromosomes in *CDKA;1^{T161D}* then decondense at diplotene and recondense toward diakinesis with a major difference being the appearance of 10 univalents instead of five bivalents (Fig 1D d and k), which is the result of an achiasmatic meiosis (no bivalents found in nine out of nine meiocytes analyzed). These univalents are rod shaped and often show fuzzy borders that may indicate problems in chromosome condensation.

The absence of synapsis and chiasmata can have several reasons, with one of the potentially earliest causes being the absence of SPO11-induced DSBs. However, the DSB repair recombinase DMC1 was localized correctly onto chromosomes with no significant reduction of foci, i.e., 138.5 ± 9.8 in *CDKA;1^{T161D}* (*n* = 10) versus 169.9 ± 15.7 (*n* = 7) in WT (*P* = 0.09, two-tailed *t*-test). This suggested that DSBs are formed along the chromosome axis and that the achiasmatic meiosis in *CDKA;1^{T161D}* results from defects in later steps of meiosis (Fig EV1F). The formation of DSBs was corroborated by the finding that a double mutant of *CDKA;1^{T161D}* with *rad51*, which is required for DSB repair, showed chromosome fragmentation (44 out of 45 meiocytes analyzed) similar to the *rad51* single mutant (39 out of 39 meiocytes; Fig EV1G). Therefore, we conclude that DSB processing, at least up to the loading of DMC1, is functional in *CDKA;1^{T161D}*. With this, we conclude that the phenotype of the hypomorphic *CDKA;1^{T161D}* mutants manifests after the meiotic DSB formation and initiation of repair but before synapsis.

Meiotic progression in *cdka;1* hypomorphic mutants is highly disturbed during meiotic stages after pachytene indicating additional roles of CDKA;1 in meiosis (Fig 1D j–n). At least a part of the cells give rise to interkinesis-like stages where two or more daughter nuclei are separated by a clear organelle band (Fig 1D l and m; 19%; *n* = 39). In such nuclei, up to 10 partially decondensed chromosomes are visible in two or more loosely organized groups, or as single chromosomes (Fig 1D l–n). A clear second meiotic division has not been observed in any cell (*n* = 206), and a phragmoplast occasionally becomes visible within the organelle band at interkinesis (in eight out of 39 cells), indicating that cytokinesis already begins at this stage (Fig 1D m). Taken together, these data suggest that CDKA;1 is an important regulator

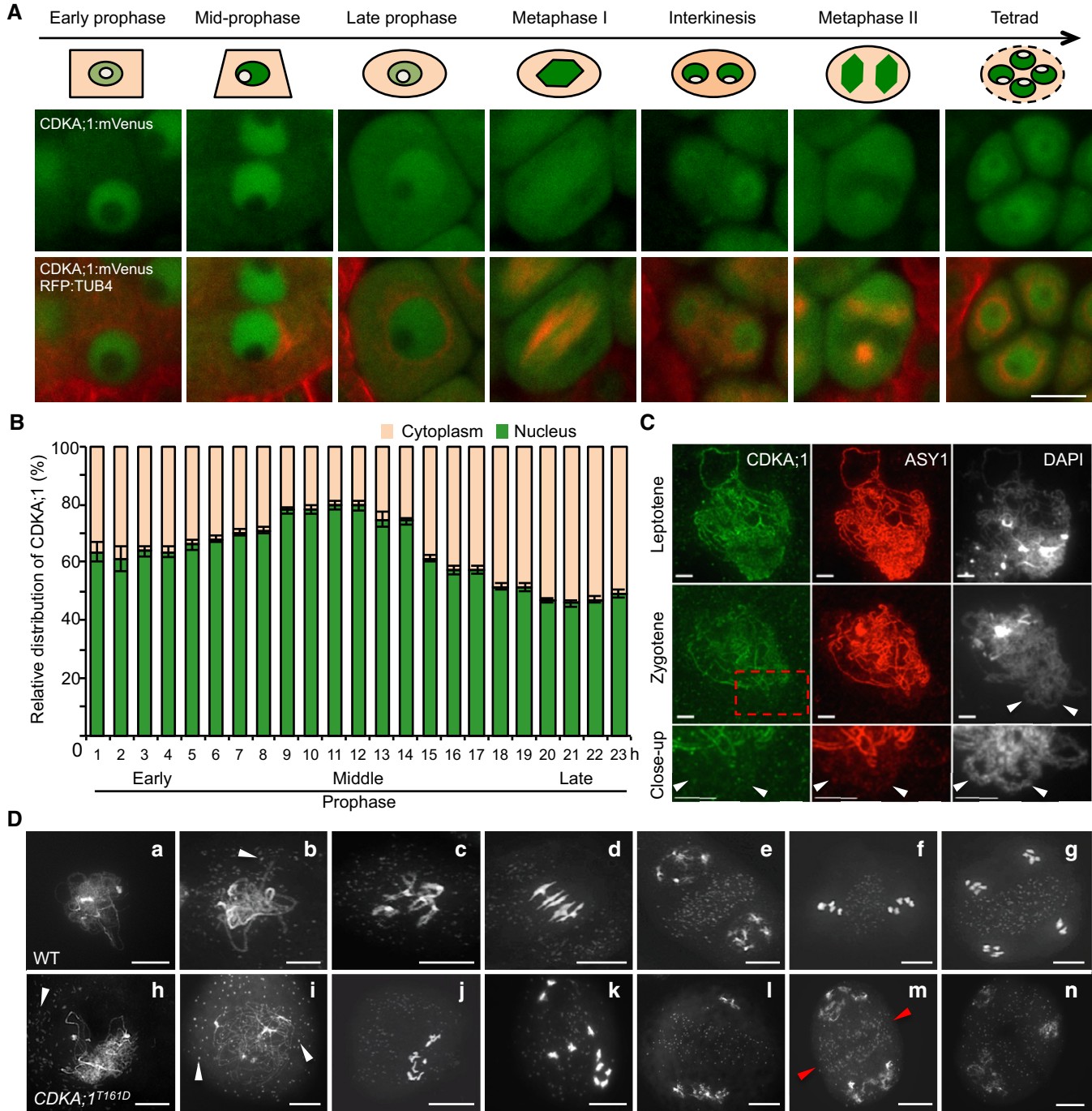

**Figure 1. Changes in CDKA;1 distribution and meiotic defects in hypomorphic *cdka;1* mutants in male meiocytes.**

A   Confocal laser scanning micrographs showing the localization of a functional CDKA;1:mVenus fusion protein in the wild type (WT) and cartoons on top highlighting the changes in abundance of CDKA;1:mVenus in the nucleus and cytoplasm during the course of meiosis. The region colored in beige represents the cytoplasm, in green the nucleoplasm, and in white the nucleolus. Scale bar: 10 μm.

B   Quantitative analysis of the signal distribution of the nuclear versus cytoplasmic fraction of CDKA;1:mVenus during prophase I of meiosis as revealed by live cell imaging (Movie EV1). Twenty cells at each time point were used for the analysis. Error bars represent mean ± SD, and two biological replicates were performed.

C   Immunolocalization of CDKA;1 (green) and ASY1 (red) on spread chromosomes in leptotene and zygotene of wild-type plants expressing a functional $PRO_{CDKA;1}$:CDKA;1:Strep construct. The last lane shows a magnification of the region marked by the red rectangle. Arrowheads indicate synapsed regions of homologous chromosomes where CDKA;1 is no longer present. Scale bar: 5 μm.

D   Chromosome spread analysis of the wild type and the hypomorphic *cdka;1* mutant *CDKA;1^{T161D}*. (a, h) zygotene or zygotene-like stages; (b, i) pachytene or pachytene-like stages; (c, j, k) diakinesis or diakinesis-like stages; (d) metaphase I; (e, i, m, n) end of meiosis I with two (e, m) or three (i) pools of chromosomes; (f) metaphase II; and (g) tetrad. Red arrowheads indicate the initiated formation of a phragmoplast. White arrowheads depict mitochondria. Scale bars: 10 μm.

of meiosis especially for chromosome synapsis and bivalent formation.

## Phosphorylation of ASY1 by CDKA;1 promotes its recruitment to the chromosome axis

Since in particular chromosome synapsis was affected in the weak loss-of-function *cdka;1* mutants, we searched for possible phosphorylation targets of CDKA;1 involved in early chromosome engagement. Several meiotic regulators in yeast have been found to contain [S/T]P Cdk consensus phosphorylation sites (Zhu *et al*, 2010). Many of these regulators have homologs in *Arabidopsis* also harboring Cdk consensus sites.

At the top of our list of putative CDKA;1 substrates was the *Arabidopsis* Hop1 homolog ASY1, especially also since *asy1* mutants are known to be asynaptic, hence partially resembling the phenotype of the hypomorphic *cdka;1* mutants (Armstrong, 2002). Moreover, a previous study identified the ASY1 ortholog of Brassica oleracea as a potential *in vivo* ATM/ATR and CDK phosphorylation target (Osman *et al*, 2017). In addition, Hop1 was found to be phosphorylated by Cdc28 in an *in vitro* screen for Cdk substrates in budding yeast (Ubersax *et al*, 2003), but the functional importance of the phosphorylation in both *Brassica* and yeast has remained unknown.

The above-mentioned spatiotemporal co-localization of ASY1 with CDKA;1 on chromosomes revealed by immunolocalization is consistent with the idea that ASY1 could be a phosphorylation target of CDKA;1 (Fig 1C). To further test this, we generated two functional reporters for *ASY1* (*PRO*$_{ASY1}$:*ASY1:GFP* and *PRO*$_{ASY1}$:*ASY1: RFP*), which both restored a wild type-like meiotic program when expressed in homozygous *asy1* mutants (Appendix Fig S1A and C). As expected, and confirming our above-presented and previous immuno-detection studies (Ferdous *et al*, 2012; Lambing *et al*, 2015), ASY1 localizes to the chromosome axis at leptotene and is depleted during zygotene when the synaptonemal complex is formed as revealed by the concomitant analysis of ASY1:RFP together with a *PRO*$_{ZYP1B}$:*ZYP1B:GFP* reporter (Figs 2A and EV2A).

To explore a possible regulation of ASY1 by CDKA;1, we introgressed the ASY1:GFP reporter into the weak *cdka;1* loss-of-function allele *CDKA;1*$^{T161D}$. In wild-type male meiocytes at late G2, numerous foci and short stretches of ASY1 signal were present (*n* = 18 out of 20 male meiocytes analyzed). The meiotic stage was determined by four morphological criteria: The squared cell shape of meiocytes, the centered position of the nucleolus, the chromosome axis being labeled by a previously generated functional *ASY3* reporter (*PRO*$_{ASY3}$:*ASY3:RFP*), and the finding that tapetum cells were still single-nucleated (Wang *et al*, 2004; Yang *et al*, 2006; Stronghill *et al*, 2014; Prusicki *et al*, 2019). In contrast, only a diffuse ASY1: GFP signal without any foci could be detected in the nuclei of meiocytes of *CDKA;1*$^{T161D}$ plants (25 out of 25) at a moment when ASY3 forms foci and short stretches (Fig 2A). This diffuse signal of ASY1: GFP in *CDKA;1*$^{T161D}$ persisted until early leptotene (19 out of 21), as judged by the beginning of the migration of the nucleolus toward one side of the nucleus and the appearance of ASY3 in threads. At this stage, a linear ASY1 signal co-localizes with ASY3 along chromosomes in the wild type (23 out 23; Fig 2A). In late leptotene, as seen by docking of the nucleolus to one side of the nucleus, the ASY1:GFP signal in *CDKA;1*$^{T161D}$ (30 out of 30) was found to associate with chromosomes indistinguishable from the wild type (28 out

28), indicating a delayed assembly of ASY1 on chromosomes in *CDKA;1*$^{T161D}$ (Fig 2A).

To test whether ASY1 can be directly phosphorylated by CDKA;1, we performed *in vitro* kinase assays. To this end, we expressed and purified ASY1 from baculovirus-infected insect cells and incubated it with three meiotic CDK–cyclin complexes. This revealed that ASY1 is phosphorylated by CDKA;1-SDS and CDKA;1-TAM but not by CDKA;1-CYCA3;1 *in vitro* (Fig 2B). Since the kinase reaction without added CDK–cyclin complexes showed background phosphorylation, likely due to co-purification of kinases from insect cells, we expressed ASY1 in *Escherichia coli* and subjected the purified protein to CDKA;1-SDS complexes. Subsequent mass spectrometry analyses showed that two sites (T142 and T535) out of the five CDKA;1 consensus phosphorylation sites in ASY1 are targeted by CDKA;1-SDS; in this case, no phosphorylated peptides were found in the reactions without CDKA;1 (Fig 3A and Appendix Fig S2A).

To address the relevance of the phosphorylation sites *in vivo*, we then generated different non-phosphorylatable and phosphorylation-mimicking variants of these five CDKA;1 consensus phosphorylation sites based on the ASY1:GFP construct. These constructs were then introduced into *asy1* mutants harboring the *ASY3:RFP* reporter (*PRO*$_{ASY3}$:*ASY3:RFP*; Table 1 and Fig EV3A). ASY3 is known to be recruited to the chromosome axis prior to ASY1 and present on chromosomes from early leptotene until pachytene (Ferdous *et al*, 2012). Consistent with its chromosomal loading being independent of ASY1, the expression and localization of ASY3 was unaffected in plants harboring different ASY1 variants and hence was used in the following as a marker for staging of meiosis (Fig 3B).

Similar to wild-type ASY1, the triple non-phosphorylatable mutant (ASY1$^{3V}$), i.e., ASY1 harboring the three amino acid substitutions T365V, S382V, T535V, and even the quadruple non-phosphorylatable mutant ASY1$^{4V}$ (T184V, T365V, S382V, T535V) fully complemented the defects of *asy1*, e.g., pollen abortion, short silique length, and reduced seed set (Table 1 and Fig EV3B–F). Matching their complementing functionality, ASY1$^{3V}$:GFP and ASY1$^{4V}$:GFP localized on chromosomes similar to ASY1:GFP that associated with chromosomes at leptotene and progressively dissociated again upon synapsis during zygotene and pachytene, while ASY3 still remained localized to the chromosomes (Fig EV2B).

In contrast, the quintuple non-phosphorylatable mutant ASY1$^{5V}$ (T142V, T184V, T365V, S382V, T535V) did not properly localize to chromosomes (Fig EV2B). Resembling *asy1* null mutants, no clear chromosomal threads were observed in ASY1$^{5V}$ plants, which were also strongly reduced in fertility (Table 1, Figs EV2B and EV3B–F). This result suggested that the *in vitro* identified CDKA;1 phosphorylation site T142 in the HORMA domain is crucial for the chromosome association of ASY1. In support of this hypothesis, we found that the single non-phosphorylatable mutant (ASY1$^{T142V}$ in *asy1*) only partially complemented *asy1*, and in contrast with the wild-type version of ASY1, ASY1$^{T142V}$:GFP showed compromised chromosome association during leptotene and exhibited a diffuse and nucleoplasmic signal (Fig 3B–D, Table 1, and Fig EV3B–F). We also frequently observed only partially synapsed homologous chromosomes in ASY1$^{T142V}$ plants (Fig 3E and Appendix Fig S3).

An exchange of T142 to serine (ASY1$^{T142S}$), that maintains the CDKA;1 phosphorylation site, did not result in compromised ASY1 function and expression of this construct fully rescued *asy1* mutants (Figs 3B and EV3B–F). Moreover, the expression of the

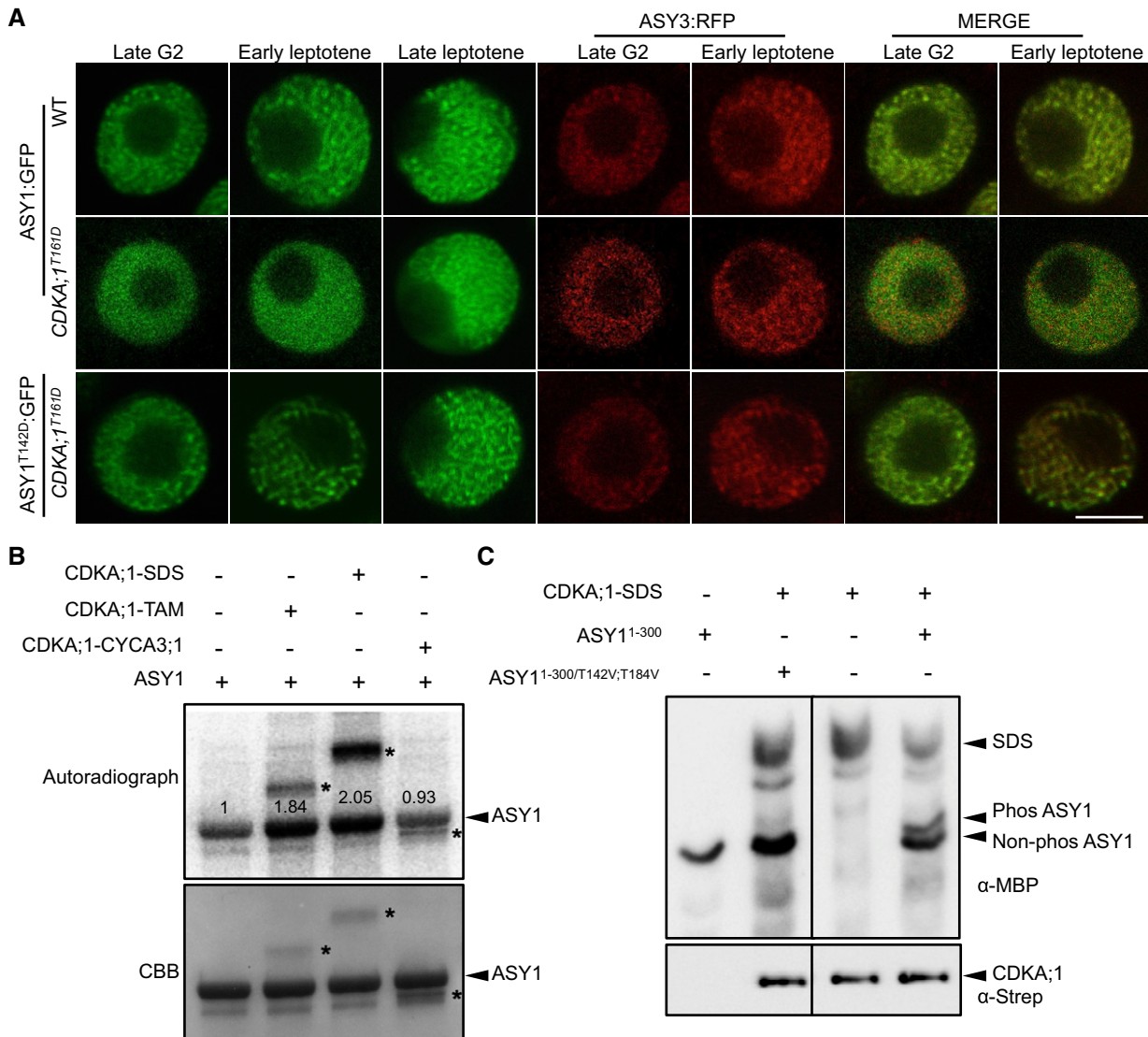

**Figure 2. ASY1 is a phosphorylation target of CDKA;1.**

A ASY1:GFP and ASY1$^{T142D}$:GFP localization in late G2 and leptotene of male meiocytes of the wild-type and *CDKA;1$^{T161D}$* mutants. ASY3:RFP, highlighting chromosomes, was used as a marker for the staging of meiosis. Scale bar: 5 μm.

B Kinase assays of CDKA;1-SDS, CDKA;1-TAM, and CDKA;1-CYCA3;1 complexes using ASY1 purified from baculovirus-infected insect cells as a substrate. The upper panel shows the autoradiograph. The control reaction without CDKA;1–cyclin complex indicates a background activity co-purified from insect cells. The lower panel indicates protein loading by Coomassie Brilliant Blue (CBB) staining. Arrowheads indicate ASY1 proteins, and asterisks depict the relevant cyclin used which also gets phosphorylated in the assay. Numbers indicate the relative intensities of ASY1 bands.

C The upper panel shows a phos-tag gel analysis of ASY1$^{1–300}$ and ASY1$^{1–300/T142V;T184V}$ with and without CDKA;1-SDS kinase complexes using an anti-MBP antibody. The lower panel denotes loading of CDKA;1 using an anti-Strep antibody. Arrowheads represent the proteins as indicated.

Source data are available online for this figure.

phosphorylation-mimicking variant ASY1$^{T142D}$ fully restored meiosis and fertility of *asy1* mutants indicating that most likely the charge and not the structure of the amino acid at position 142 is important for ASY1 function (Figs 3B and EV3B–F). Furthermore, the delayed assembly of ASY1 in *CDKA;1$^{T161D}$* was reverted to a wild-type pattern when the phosphorylation-mimicking mutation ASY1$^{T142D}$ was expressed in *CDKA;1$^{T161D}$* (Fig 2A).

Exploring the regulation of ASY1 phosphorylation further, we found that the double non-phosphorylatable mutant T142V, T184V (*ASY1$^{T142V;T184V}$*) enhanced the *ASY1$^{T142V}$* mutant phenotype and was indistinguishable from *asy1* indicating a complete loss of function reminiscent of *ASY1$^{5V}$* (Figs 3B and EV3B–F, and Appendix Fig S7). Consistently, the N-terminal half of ASY1 in which the phosphorylation sites T142 and T184 were mutated (ASY1$^{1-300/T142V;T184V}$) was no longer phosphorylated by a CDKA;1-SDS complex *in vitro* confirming their specificity as CDKA;1 phosphorylation sites (Fig 2C).

Since no obvious localization defects, especially in leptotene, and no mutant phenotype were found in *asy1* mutants expressing the

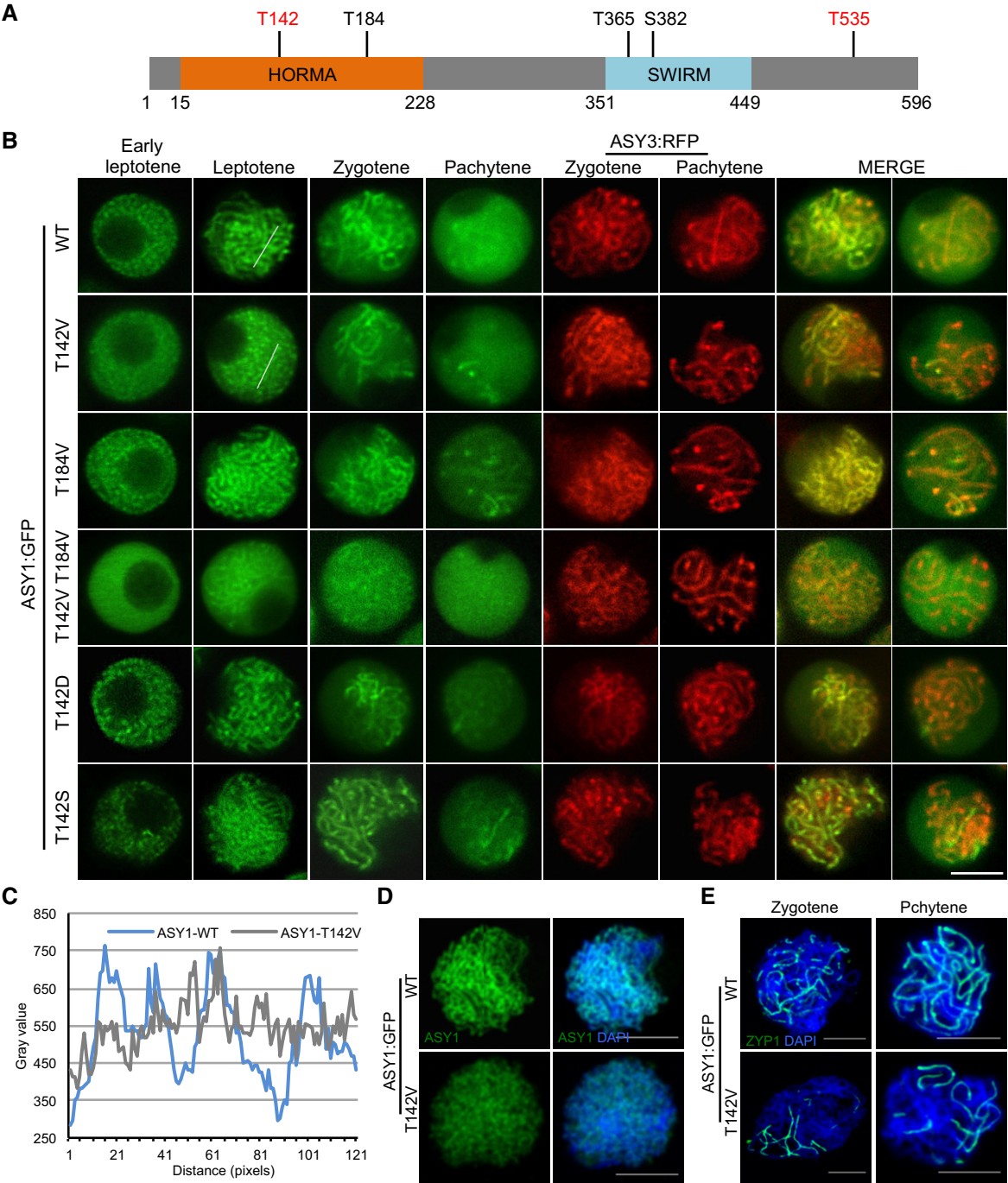

**Figure 3. Phosphorylation of ASY1 is essential for its chromosomal localization.**

A  Schematic representation of ASY1 with the five predicted consensus Cdk phosphorylation sites. The sites found to be phosphorylated *in vitro* by CDKA;1-SDS complexes are highlighted in red (Appendix Fig S4A).

B  Localization patterns of different ASY1:GFP variants together with ASY3:RFP (for staging of zygotene and pachytene) in a *asy1* mutant background during prophase I. Scale bar: 5 μm.

C  Signal distribution profiles of ASY1:GFP and ASY1^T142V:GFP at leptotene as shown in (B). The regions used for analysis are highlighted by white lines in respective panels in (B). The many small peaks with low amplitude in *ASY1^T142V:GFP* indicate diffused localization as opposed to the clear peaks seen in the wild type.

D, E  Immunolocalization of ASY1 (D) and ZYP1 (E) in *ASY1:GFP* (*asy1*) and *ASY1^T142V:GFP* (*asy1*) plants using anti-GFP and anti-ZYP1 antibodies, respectively. DNA was stained with DAPI (blue). Scale bars: 5 μm.

**Table 1. Summary of the phenotypic analysis of ASY1 variants.**

| Construct | Chromosome association | Background | Seed/silique | Pollen viability (%) |
|---|---|---|---|---|
| – | – | Wild type | $58.35 \pm 1.75^a$ | $99.32 \pm 0.49^a$ |
| – | – | *asy1* | $9 \pm 1.2^b$ | $55.57 \pm 2.55^b$ |
| ASY1 | Correct | *asy1* | $58.75 \pm 2.32^a$ | $99.26 \pm 0.63^a$ |
| ASY1$^{T142V}$ | Compromised | *asy1* | $41 \pm 2.5^c$ | $81.87 \pm 2.35^c$ |
| ASY1$^{T184V}$ | Correct | *asy1* | $57.78 \pm 2.5^a$ | $99.24 \pm 0.27^a$ |
| ASY1$^{T142V;T184V}$ | Largely lost | *asy1* | $9.78 \pm 1.8^b$ | $59.72 \pm 2.27^b$ |
| ASY1$^{3V}$ | Correct | *asy1* | $57.75 \pm 1.83^a$ | $99.38 \pm 0.4^a$ |
| ASY1$^{4V}$ | Correct | *asy1* | $58.15 \pm 1.96^a$ | $99.04 \pm 0.25^a$ |
| ASY1$^{5V}$ | Largely lost | *asy1* | $9 \pm 1.41^b$ | $55.86 \pm 3.57^b$ |
| ASY1$^{T142D}$ | Correct | *asy1* | $57.95 \pm 2.3^a$ | $99.21 \pm 0.23^a$ |
| ASY1$^{T142S}$ | Correct | *asy1* | $56 \pm 2.96^a$ | $98.19 \pm 0.9^a$ |

The level of significance ($P < 0.05$) is indicated by different letters between the wild-type and ASY1 variants as determined by the one-way ANOVA followed by Turkey's test.

single non-phosphorylatable mutant *ASY1$^{T184V}$* (Figs 3B and EV3B–F), we conclude that T142 in the HORMA domain is the major site of ASY1 phosphorylation regulation with the site T184 likely having an ancillary role.

### Phosphorylation of ASY1 increases its binding affinity with ASY3

The failure of the double non-phosphorylatable mutant protein ASY1$^{T142V;T184V}$ to associate with chromosomes is reminiscent of the localization defects of ASY1 in *asy3* mutants (Figs 3B and EV2C; Ferdous *et al*, 2012). Therefore, we reasoned that the phosphorylation of ASY1 may control its interaction with ASY3. The first 300 amino acids of ASY1 (ASY1$^{1–300}$), which include the HORMA domain, essential for the protein–protein interaction of Hop1 with Red1 (Muniyappa *et al*, 2014; Rosenberg & Corbett, 2015), were found to interact with ASY3 in a yeast two-hybrid assay consistent with earlier results (Ferdous *et al*, 2012). While no obvious effect of ASY1$^{1–300/T184V}$ on the interaction capacities with ASY3 was observed, we found that the binding of ASY1$^{1–300/T142V}$ to ASY3 was strongly decreased, yet not fully abolished, since yeast cells harboring ASY1$^{1–300/T142V}$ and ASY3 cannot grow on the stringent selection media (without histidine and adenine) but do survive on the less stringent media (without histidine; Fig 4A). The interaction with ASY3 was even further reduced in the ASY1$^{1–300/T142V;T184V}$ variant (Fig 4A). Conversely, the phosphorylation site exchange mutant ASY1$^{1–300/T142S}$ and the phosphorylation-mimicking mutant ASY1$^{1–300/T142D}$ interacted with ASY3 to a similar extent as the non-mutated version of ASY1 (Fig 4A). These findings were not due to protein expression levels since we found that non-phosphorylatable mutant versions of ASY1 were even more abundantly present in yeast cells than the non-mutated version (Appendix Fig S2C). These results also suggested that ASY1 was phosphorylated in yeast cells, likely on residue T142. Indeed, the phosphorylation of ASY1 in yeast was confirmed by phos-tag SDS–PAGE (Appendix Fig S2D).

The importance of T142 phosphorylation in ASY1 for the interaction with ASY3 was confirmed by GST pull down assay using recombinant proteins purified from *E. coli*. Similar to the results

from yeast two-hybrid assay, we found that the non-phosphorylated ASY1 (ASY1$^{1–300}$ and ASY1$^{1–300/T142V;T184V}$) had only a residual interaction capacity with ASY3. However, the phosphorylation-mimicking version ASY1$^{1–300/T142D}$ showed enhanced affinity toward ASY3 (Fig 4B and C).

Finally, we addressed whether and if so to what degree the altered interaction of ASY1$^{1–300/T142V}$ with ASY3 in our yeast two-hybrid experiment depends on the exchanged amino acid, i.e., Val, itself. To this end, we tested additional ASY1 variants in which we substituted T142 and T184 with Gly and Ala. Consistent with the Val substitution at T142, we found that the mutations of both ASY1$^{1–300/T142A}$ and ASY1$^{1–300/T142G}$ strongly reduced the interaction with ASY3 (Fig EV4A). The mutation of ASY1$^{1–300/T142A;T184A}$ further reduced the interaction of ASY1$^{1–300/T142A}$ with ASY3 similarly to the ASY1$^{1–300/T142V;T184V}$ mutant (Fig EV4A). These results were not attributed to protein expression levels since we found that all non-phosphorylatable mutant versions of ASY1 were not less abundant in yeast cells than the wild-type version (Appendix Fig S2E). These findings show that a reduced interaction between ASY1 and ASY3 in a yeast two-hybrid system does not depend on a specific amino acid used for substitution and corroborate that the phosphorylation of ASY1 at T142 is important for its binding to ASY3.

Notably, while ASY1$^{1–300/T184V}$ does not show any obvious reduction in binding with ASY3, the substitution of T184 to A (ASY1$^{1–300/T184A}$) largely reduced the interaction, and ASY1$^{1-300/T184G}$ did not interact at all with ASY3 in our assays anymore (Fig EV4A). Since we did not find T184 to be phosphorylated *in vitro*, we cannot judge at the moment whether T184 is structurally a very important position and does not tolerate small amino acids and/or whether T184 is, possibly very transiently, phosphorylated *in vivo*.

### Phosphorylation of ASY1 counteracts the action of PCH2 in early but not late prophase

For the synaptonemal complex to be formed, ASY1 has to be depleted from synaptic regions at zygotene mediated by the conserved AAA$^+$ ATPase PCH2 (Ferdous *et al*, 2012; Lambing *et al*,

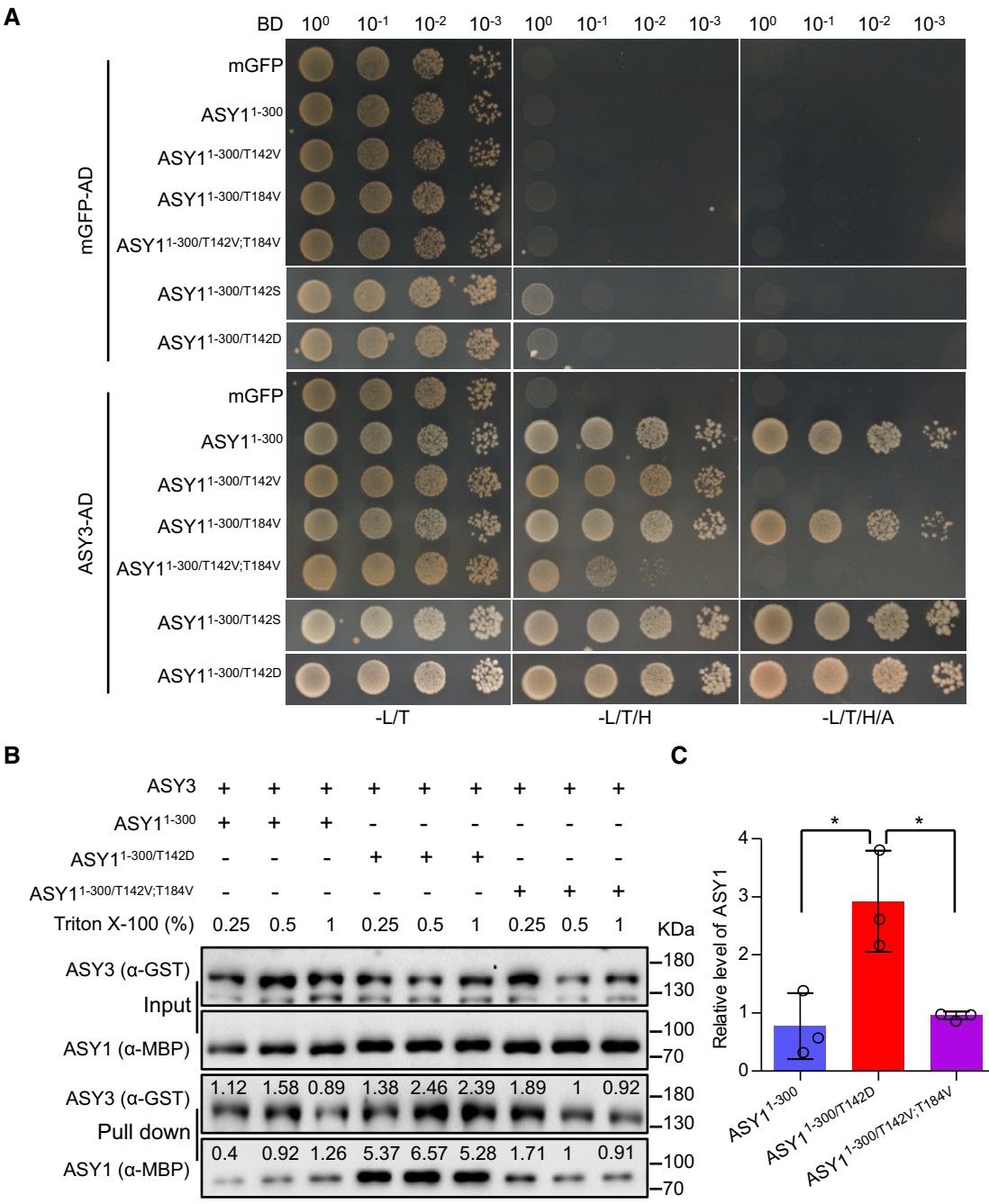

**Figure 4. A negative charge at T142 in the HORMA domain of ASY1 promotes its interaction with ASY3.**

A   Yeast two-hybrid interaction assays of ASY3 with different ASY1 variants. Monomeric GFP (mGFP) fused with AD (activating domain) and BD (binding domain) were used as controls. Yeast cells harboring both the AD and BD plasmids were grown on synthetic medium supplied with glucose in the absence of Leu and Trp (-L/T, left panel), on synthetic dropout (SD) medium in the absence of Leu, Trp, and His (-L/T/H, middle panel), and on SD medium in the absence of Leu, Trp, His and Ade (-L/T/H/A, right panel). Yeast cells were incubated until OD$_{600}$ = 1 and then diluted 10-, 100-, and 1,000-fold for the assays.

B   GST pull down of ASY3 with different ASY1 variants. The numbers above the bands show the relative intensity of the bands. The input and pull down fractions were analyzed by immuno-blotting with the anti-GST (ASY3) and anti-MBP (ASY1) antibodies.

C   Quantification of the pull down fractions of ASY1 as shown in (B). The band intensity in the pull down of ASY1$^{1–300/T142V;T184V}$ at a Triton X-100 concentration of 0.5% was defined as 1. The relative amount of ASY1 in the pull down fractions was normalized by the band intensity of the pulled down ASY3 fraction. The average band intensity of ASY1 at different concentrations of Triton X-100 used was plotted. Asterisks indicate significant difference (two-tailed *t*-test, $P < 0.05$). Error bars represent mean ± SD, and two biological replicates were performed.

2015). Therefore, we asked whether the phosphorylation status of ASY1 also affects its removal by PCH2. However, the phosphorylation-mimicking version ASY1$^{T142D}$ was equally well depleted from chromosomes as the non-mutated version of ASY1 (Fig 3B). Conversely, we introduced the non-phosphorylatable version ASY1$^{T142V;T184V}$ into *pch2* mutants to ask whether the loss of the chromosomal association of the ASY1$^{T142V;T184V}$ was affected by PCH2.

Strikingly, while ASY1$^{T142V;T184V}$ could not properly localize to chromosomes in both *asy1* mutant and a wild-type background (see above, Figs 3B and 5A), the localization pattern of ASY1$^{T142V;T184V}$:GFP in *pch2* was nearly identical to the pattern of the non-mutated version of ASY1 in leptotene (Fig 5A). This observation suggests a so far not recognized function of PCH2 in counteracting the recruitment of ASY1 to ASY3 in leptotene when ASY1 needs to assemble on the chromosomes (see below). This early function of PCH2 for the regulation of the chromosome assembly of ASY1 at leptotene was further corroborated by the finding that although ASY1$^{T142V;T184V}$:GFP could not rescue the fertility reduction of *asy1* mutants, it largely complemented the fertility of *asy1 pch2* double mutants to the level of *pch2* single mutants (Fig 5B). This result also suggested that ASY1$^{T142V;T184V}$:GFP is largely functional as long as it can be localized on chromosomes. The finding that ASY1$^{T142V;T184V}$ in a *pch2* mutant background stays tightly associated with the chromosomes at both zygotene and pachytene when ASY1 in wild-type plants is already largely removed from the synaptic chromosomes, underlines the key role of PCH2 for the late release, which we conclude is independent of the CDKA;1-dependent phosphorylation status of ASY1 (Figs 3B and 5A).

To further explore the new finding of an early function of PCH2, we generated a functional genomic reporter line for *PCH2* (*PRO$_{PCH2}$: PCH2:GFP*) which revealed that PCH2 is already present in male meiocytes from pre-meiosis throughout prophase I (Fig 5D, Appendix S1B and D). This observation implies the necessity of a mechanism for counteracting the releasing force of PCH2 on ASY1 at early leptotene, which we speculate to be the here-discovered phosphorylation of ASY1.

To elaborate on a possible early function of PCH2, we introduced the non-mutated functional *ASY1:GFP* reporter into *pch2* mutants. While ASY1:GFP is exclusively localized to the nucleus and chromosomes in a wild-type background, we found that the same reporter was not only present in the nucleus but also strongly accumulated in the cytoplasm in *pch2* mutants (Fig 5C). Revisiting the non-phosphorylatable mutant localization ASY1$^{T142V;T184V}$:GFP in *pch2*, we also observed that it accumulates cytoplasmically (Fig 5C). Interestingly, we noted that the signal intensities of both ASY1:GFP and ASY1$^{T142V;T184V}$:GFP in the nucleus of *pch2* mutants appeared to be weaker than that of ASY1:GFP in the wild type (Fig 5C). Thus, we conclude that PCH2 directly or indirectly facilitates the nuclear accumulation of ASY1 in early meiosis, a function that is consistent with the presence of PCH2 in the cytoplasm at that time (Fig 5D).

Taken together, these observations suggest that PCH2 has at least three, possibly interconnected functions. In early leptotene, it promotes the release of the non-phosphorylated ASY1 from chromosomes and ASY1 phosphorylation in the HORMA domain antagonizes this PCH2 activity by increasing the binding affinity of ASY1 with ASY3. At the same time, PCH2 helps ASY1 to accumulate in the nucleus. Later in zygotene and pachytene, PCH2 removes ASY1,

as shown in previous publications, in a fashion that appears to not depend on its phosphorylation (Lambing *et al*, 2015).

## Self-assembly of ASY1 through its C-terminal closure motif is affected by the phosphorylation in the HORMA domain

The chromosomal localization of the meiotic HORMA domain proteins (HORMADs) including the budding yeast Hop1, mammalian HORMAD1 and HORMAD2, and *Caenorhabditis elegans* HORMADs (HTP-1, HTP-2, HTP-3, and HIM3) was recently shown to depend on at least two mechanisms, the initial recruitment by its binding partners such as Red1 in yeast, and the putative self-assembly through its N-terminal HORMA domain-C-terminal closure motif interactions (Smith & Roeder, 1997; Wojtasz *et al*, 2009; Kim *et al*, 2014; West *et al*, 2018). Hence, we asked whether the phosphorylation by CDKA;1 would also affect a possible self-assembly mechanism of ASY1.

To explore this possibility, we first tested whether the self-assembly is also conserved in *Arabidopsis* by using yeast two-hybrid assays. We found that ASY1 binds to itself and mapped this interaction to the HORMA domain of ASY1 making contact with the very C-terminus of ASY1 (residues 571–596), strongly suggesting that ASY1 likely possesses a C-terminal closure motif as its orthologs in yeast, *C. elegans*, and mammals (Fig 6A and B). While this work was in progress, West *et al* (2019) also independently identified the closure motif of ASY1 as being located in the same region as here revealed by us. Deletion of the closure motif of ASY1 in the *ASY1: GFP* reporter construct (ASY1$^{1-570}$:GFP) almost abolished its chromosome association, indicating the necessity of the closure motif for its correct localization pattern (Fig 5C). At the same time, we also observed that ASY1$^{1-570}$:GFP accumulated in the cytoplasm demonstrating that the nuclear targeting of ASY1 is also compromised in this version. Next, we asked whether the compromised chromosome association of ASY1$^{1-570}$:GFP depends on PCH2. Remarkably, the chromosome localization of ASY1$^{1-570}$:GFP was largely recovered (Fig 5C), when the *ASY1$^{1-570}$:GFP* reporter was introduced into *pch2* mutant. This suggests that the closure motif is also important for antagonizing the releasing force of PCH2, presumably via the self-oligomerization during chromosome axis formation.

We also noticed in our yeast two-hybrid assays that the full-length ASY1 could not interact with ASY3 (Fig 6B). This is consistent with previous studies that show that full-length Hop1 has a very low affinity toward Red1 *in vitro* (West *et al*, 2018). However, strong interaction with ASY3 was found when the closure motif was depleted (ASY1$^{1-570}$; Fig 6B).

Finally, we tested the interaction of the different mutant variants of the ASY1 HORMA domain with the above-identified closure motif and found that the affinity of ASY1$^{1-300/T142V}$ and ASY1$^{1-300/T142V;T184V}$ to the closure motif was dramatically reduced. Conversely, the phosphorylation-mimicking version ASY1$^{T142D}$ showed higher interaction strength despite a slight decrease compared to that of the non-mutated ASY1 version (Fig 6C). These data suggest that the phosphorylation of the ASY1 HORMA domain regulates its chromosomal assembly not only by enhancing the affinity to ASY3 but also by promoting the potential self-assembly along the chromosomes. Thus, the phosphorylation of ASY1 by CDKA;1 appears to represent a two-pronged mechanism for the faithful loading of ASY1 to the chromosome axis.

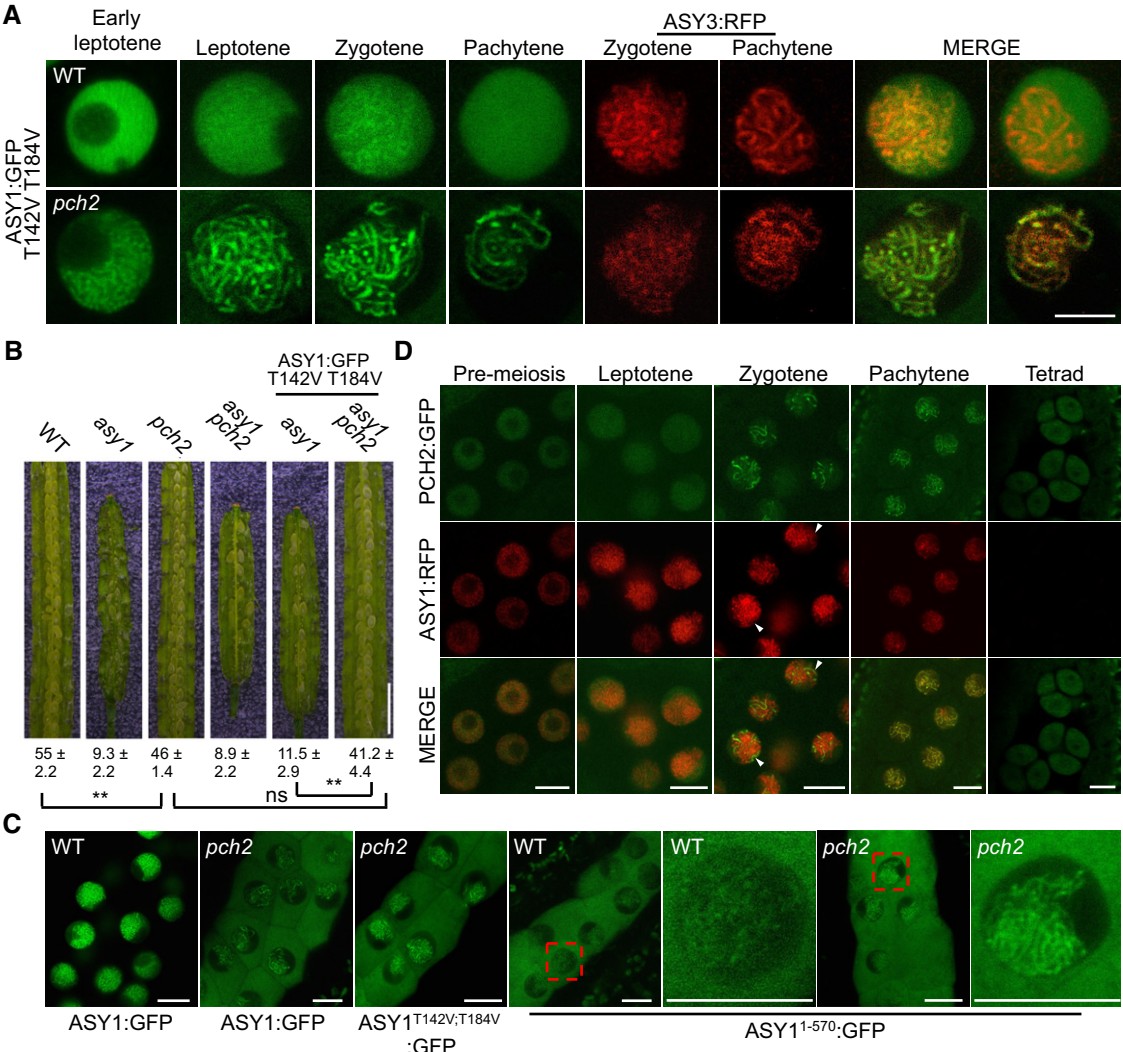

**Figure 5. Phosphorylation of ASY1 counteracts the action of PCH2 in early prophase.**

A   Localization patterns of ASY1$^{T142V;T184V}$:GFP together with ASY3-RFP in the wild-type and *pch2* mutants. Please note that images of ASY1$^{T142V;T184V}$:GFP in *pch2* mutants were taken with increased sensitivity for a better visibility. Scale bar: 5 μm.

B   Seed sets (mean ± SD, *n* = 5) of WT, *asy1*, *pch2*, *asy1 pch2*, ASY1$^{T142V;T184V}$:GFP (*asy1*), and ASY1$^{T142V;T184V}$:GFP (*asy1 pch2*) plants. Asterisks indicate significant differences (two-tailed *t*-test, *P* < 0.01), and ns depicts no significant difference. Scale bar: 2 mm.

C   Localization patterns of ASY1:GFP, ASY1$^{T142V;T184V}$:GFP, and ASY1$^{1–570}$:GFP in the wild-type (WT) and/or in *pch2* mutant plants at early prophase I. Scale bars: 5 μm.

D   Localization pattern of PCH2:GFP together with ASY1:RFP in the male meiocytes of wild type. Arrowheads indicate the chromosomal regions where the ASY1 removal was concomitant with the localization of PCH2. Scale bar: 10 μm.

Source data are available online for this figure.

# Discussion

Cdks are known to be the major driving force of cell divisions (Morgan, 1997). Due to their requirement in mitosis, the study of Cdks in meiosis is challenging in multicellular organisms since meiosis usually takes place late during embryonic or postembryonic development, i.e., after several mitotic divisions. This is exemplified by the early embryonic lethality of Cdk1 mutants that precludes a straightforward functional analysis of Cdk1 in mouse meiocytes (Santamaría *et al*, 2007). By replacing Cdk1 with Cdk2 and by using conditional Cdk1 knock out mice, it was shown that Cdk1 is indeed key for meiosis in mammalian oocytes and cannot be substituted by

Cdk2 (Satyanarayana *et al*, 2008; Adhikari *et al*, 2012). However, it is still largely not clear how Cdk1 controls meiotic progression and what the phenotypic consequences of the loss of Cdk1 activity in meiosis are at the cellular level.

Since weak loss-of-function mutants in the *Arabidopsis* Cdk1/Cdk2 homolog *CDKA;1* are viable and produce flowers containing meiocytes (Dissmeyer *et al*, 2007, 2009), they represent a unique tool to study the requirement of Cdks in meiosis of a multicellular eukaryote. Exploiting these mutants, we find that in particular chromosome synapsis and bivalent formation are affected by reduced Cdk activity. However, the identification of Cdk targets in multicellular organisms is still challenging. Especially for specific tissues

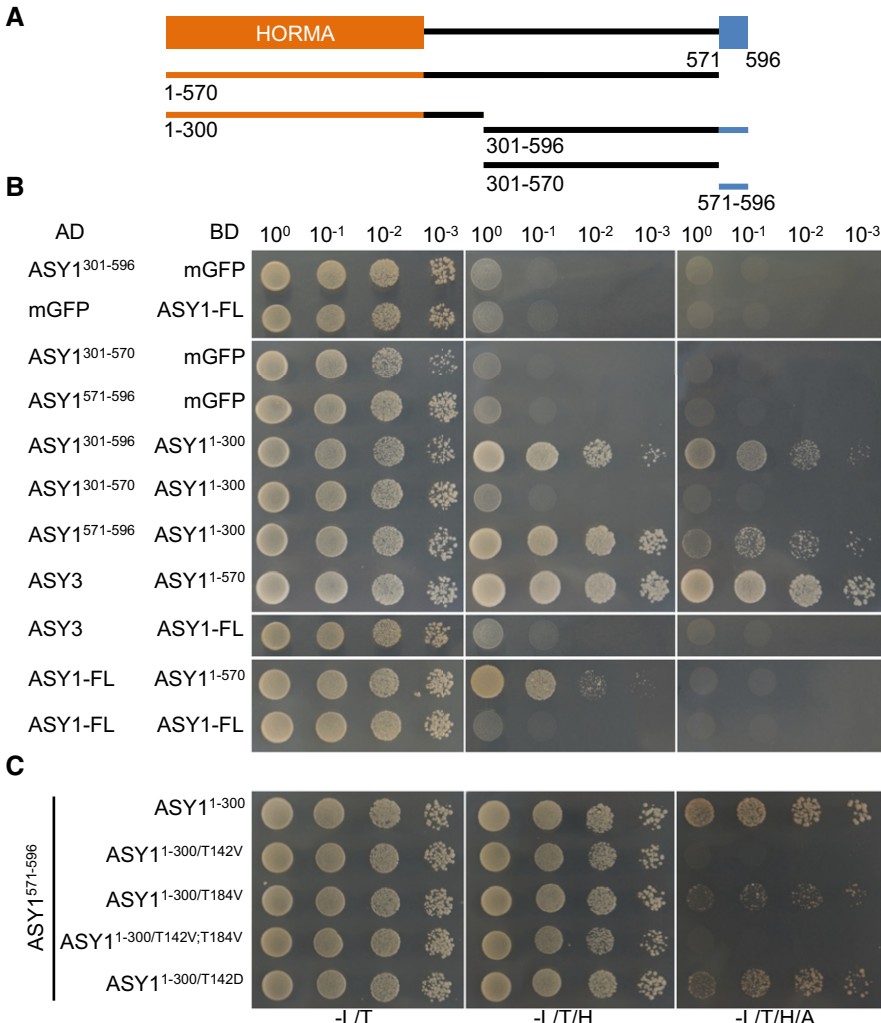

**Figure 6. Phosphorylation of ASY1 affects its self-assembly.**

A Schematic graph of ASY1 full-length protein (aa 1–596). The HORMA domain is depicted in orange, and the presumptive closure motif is highlighted in blue. The lines below indicate the constructs used for yeast two-hybrid interaction assays.
B Interaction assays of different ASY1 fragments (with and without the closure motif).
C Interaction analysis of the ASY1 closure motif (ASY1$^{571–596}$) with different ASY1 HORMA domain variants.

such as meiocytes, and/or when phosphorylated proteins are only transiently present, the power of phosphoproteomics approaches and the identification of the phosphorylation sites *in vivo* are still limited. Combining genetic, cytological, and biochemical approaches, we have accumulated evidence, suggesting that the phosphorylation of ASY1 by CDKA;1 complexes is needed for the formation of the chromosome axis in meiosis. We show that T142 is very likely the key phosphorylation-dependent regulatory site, with T184 playing an ancillary role.

So far, the ASY1 homolog Hop1 has been found to be phosphory-lated by Mec1/ATR and Tel1/ATM in budding yeast, which promotes DMC1-dependent interhomolog recombination without affecting the chromosomal association of Hop1 (Carballo *et al*, 2008). Orthologs of Hop1 in plants, e.g., ASY1 in *Arabidopsis* and PAIR2 in rice, harbor also ATM/ATR consensus phosphorylation sites ([S/T]Q), but whether an ATM/ATR-dependent

phosphorylation is functionally conserved in plants is still unclear. Given the finding that Hop1 can be also phosphorylated by Cdk complexes in budding yeast (Ubersax *et al*, 2003) and the presence of Cdk consensus phosphorylation sites in both HORMAD1 and HORMAD2 proteins of human and mouse, it is tempting to speculate that the here-revealed phosphorylation regulation of ASY1, needed for its chromosome localization, is conserved among eukaryotes.

### The role of ASY1 phosphorylation by CDKA;1

Combining our CDKA;1 localization data with the molecular and biochemical analysis of ASY1 phosphorylation, we propose a model of how CDKA;1 regulates ASY1 (Fig 7). At early prophase I, CDKA;1 changes from a distribution of approximately 40% versus 60% in the cytoplasm and nucleus to a prominently nuclear localization (~80%), likely promoted by a meiotic cyclin such as SDS. In the

nucleus, CDKA;1 phosphorylates ASY1 and by that enhances its binding affinity with ASY3. It is possible that CDKA;1 acts directly at the chromosome axis based on our immunolocalization data (Fig 1C). The related kinases Cdk2 from mammals and Cdc28 from budding yeast have both been found to localize to chromatin, too (Ashley *et al*, 2001; Zhu *et al*, 2010). However, these two kinases show a punctuate localization pattern, while CDKA;1 has more continuous appearance along chromosome axis resembling the localization of ASY3 and ASY1 itself (Ashley *et al*, 2001; Armstrong, 2002; Ferdous *et al*, 2012).

The phosphorylation of ASY1 has several consequences. First, it enhances the affinity toward ASY3 promoting the recruitment of ASY1 to the chromosome axis. Second, it antagonizes a releasing force executed by PCH2, which is already present very early in meiosis. At the same time, PCH2 promotes the nuclear accumulation of ASY1 (Fig 7).

Moreover, the phosphorylation of ASY1 likely promotes the formation of ASY1 polymers similar to the proposed Hop1/HORMADs polymers in budding yeast *Saccharomyces cerevisiae* and *C. elegans*, that is likely essential for its chromosome localization (Kim *et al*, 2014; Rosenberg & Corbett, 2015; West *et al*, 2018). HORMA domain proteins, such as Hop1, have been shown to bind to closure motifs in partner proteins (West *et al*, 2018). This interaction is stabilized by the folding of the C-terminal safety belt region of the HORMA domain protein around this binding motif from the respective partner resulting in a so-called closed state. Meiotic HORMA domain proteins such as Hop1 contain themselves closure motifs and have been shown to bind to other HORMAD molecules and by that likely leading to HORMAD polymers along the

unsynapsed chromosome axes (Kim *et al*, 2014; West *et al*, 2018). These polymers are presumably anchored by binding to cohesin and/or axis proteins such as Red1/ASY3. However, the full-length *Arabidopsis* ASY1 (ASY1$^{FL}$) showed a very low affinity toward ASY3 in our yeast two-hybrid assays (Fig 6B). Similar findings were recently reported for Hop1 and Red1 *in vitro* (West *et al*, 2018). The binding capacity of ASY1 to ASY3 was strongly enhanced when the short C-terminal region of ASY1 including the presumptive closure motif was deleted (Fig 6B). These results argue that a full-length ASY1, at least when being expressed in yeast cells, is in a closed conformation being bound by its own closure motif in the C-terminus or by the closure motif from another ASY1. However, we could not detect any interaction of ASY1$^{FL}$ to ASY1$^{FL}$ using the yeast two-hybrid assay, suggesting that ASY1 tends to fold in a closed state through binding to its own closure motif at least when being expressed in yeast (Fig 6B). Assuming that the same holds true *in planta*, one needs to postulate that there is a factor that regulates the close-to-open state switch of ASY1. Our finding that ASY1 accumulates in the cytoplasm in *pch2* mutants suggests that PCH2 could have a function in converting ASY1 from the closed to the open state and at the same time probably avoiding the premature polymerization (in the cytoplasm) and providing a pool of available and reactive ASY1.

Mapping the ASY1 protein sequence onto the structure of *C. elegans* HIM-3 (c4trkA) using Phyre2 protein folding prediction shows that the T142 residue is likely located at the N-terminus of the alpha-C helix, a position just at the terminus of the long loop between beta-5 and alpha-C (Fig EV4B; Kim *et al*, 2014). Since this loop anchors the C-terminal safety belt in place, one idea might be

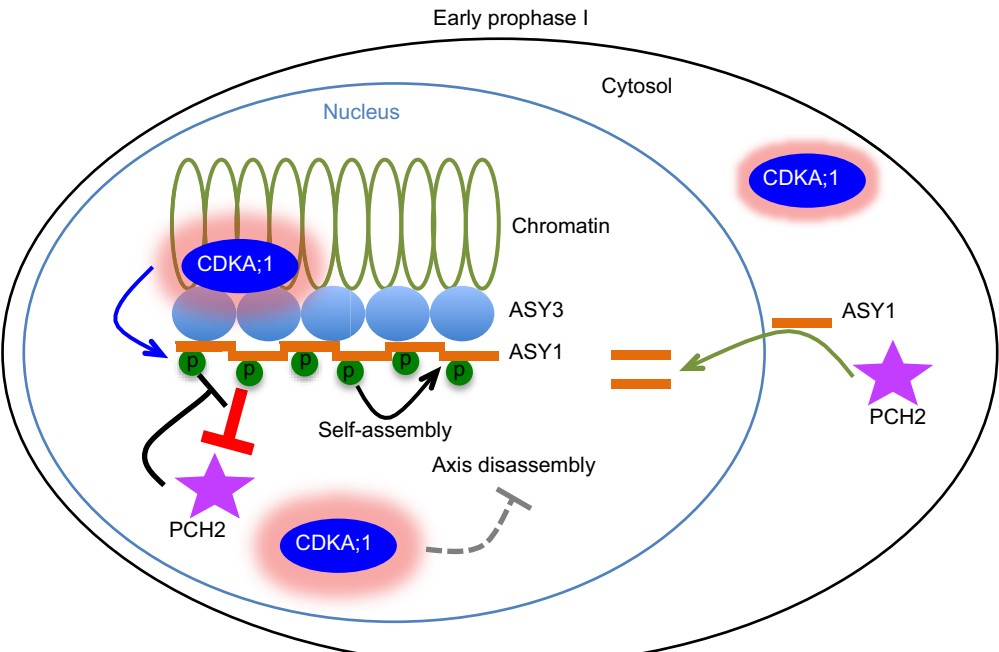

**Figure 7.  Model for the regulation of the chromosomal assembly of ASY1.**

In early prophase I, ASY1 is expressed and imported into nucleus, facilitated by PCH2. Concomitantly, CDKA;1 becomes enriched in the nucleus, localizes on chromosomes, and phosphorylates ASY1. The phosphorylation enhances the binding affinity of ASY1 to ASY3 and the self-assembly and thus, in turn, antagonizes the releasing force of PCH2. At the same time, high CDKA;1 activity in the nucleus may block other axis disassembling factors that will be activated later in synapsed regions where CDKA;1 is not present.

that the phosphorylation of T142 imparts greater flexibility to the loop, and thereby may allow the safety belt to disengage, i.e., to "open" the protein and thus, allow the closure motif binding/dissociation (Kim *et al*, 2014; West *et al*, 2018).

When homologs synapse at zygotene and pachytene, ASY1 is displaced from chromosome axes by PCH2 and this removal is essential for completing chromosome synapsis and recombination (Lambing *et al*, 2015). Concomitantly with the ASY1 removal, the nuclear levels of CDKA;1 drop and CDKA;1 is also evicted from chromatin of synapsed regions (Fig 1A–C). Whether a possible drop in CDKA;1 activity in the nucleus is relevant for the removal of ASY1 is not clear. At least the phosphorylation-mimicking mutant ASY1$^{T142D}$ can be released from chromatin indicating that the removal of ASY1 functions independently from its phosphorylation. This suggests either that an unknown regulator/cofactor of PCH2 exists, which enhances the activity of PCH2, or that PCH2 has a higher activity at synaptic regions. The latter is supported by the observation that while the PCH2 signal shows a diffuse nuclear localization before zygotene when ASY1 is assembled on the chromosome axis, it starts to accumulate specifically at the synaptic regions at zygotene coinciding with ASY1 removal. After that, PCH2 is largely present along the entire chromosomes at pachytene (Fig 5D). It is tempting to speculate that CDKA;1 might phosphorylate and by that inhibit an ASY1 disassembly factor (Fig 7). Hence, a reduction of CDKA;1 in the nucleus as seen here by live cell imaging could also throw the switch for this removal step. Although PCH2 has a Cdk phosphorylation site, it seems unlikely that PCH2 itself could be the target of this potential mechanism since we found here that at a phase of presumed high CDKA;1 activity, PCH2 is able to displace ASY1 from the chromosome axis as seen by the restoration of this interaction in a *pch2* mutant background (Fig 5A and B). On the other hand, the removal of ASY1 at zygotene may be regulated through other post-translational modifications. Consistent with this hypothesis, Osman *et al* (2017) have identified several other phosphorylation sites on ASY1, notably ATM/ATR phosphorylation sites. Thus, further work is required to understand the mechanisms of how ASY1 is removed from the chromosome axis.

### Beyond ASY1 phosphorylation

Here, we have shown that CDKA;1 works together with SDS and TAM. However, the *sds* mutant phenotype is not a subset of the phenotype of the weak loss-of-function *cdka;1* mutants as seen by the apparently correct localization of DMC1 in *cdka;1* versus the localization failure in *sds* (De Muyt *et al*, 2009). One possible explanation is that SDS can work with additional Cdks, such as CDKB1;1, which have been recently shown to function in somatic homologous recombination repair (Weimer *et al*, 2016). However, at least *in vitro* neither CDKB1;1 nor the related kinase CDKB2;2 built an active kinase complex with SDS (Harashima & Schnittger, 2012). Thus, it seems more likely that the residual Cdk activity in the hypomorphic mutants is sufficient to operate together with SDS to promote DMC1 loading/stabilization. Notably, the localization of ASY1 to chromatin is also only delayed and not completely absent in weak loss-of-function mutants.

Earlier work has already indicated that TAM, the other meiotic cyclin used in our assays, is needed to promote the timely progression through meiosis I and entry into meiosis II (d'Erfurth *et al*, 2010). At the same time, CDK-dependent phosphorylation of THREE DIVISION MUTANT 1 (TDM1) has been shown to be crucial for the exact timing of meiotic exit. Mutation of the CDK phosphorylation site in TDM1 also results in termination of meiosis after anaphase I (Cifuentes *et al*, 2016). Furthermore, the loss of the APC/C inhibitor OMISSION OF SECOND MEIOTIC DIVISION 1 (OSD1), also known as GIGAS CELL 1 (GIG), and the presumed increase in APC/C activity also caused a premature termination of meiosis after anaphase I (Iwata *et al*, 2011; Cromer *et al*, 2012). Consistently with these studies, we found that weak loss-of-function mutants of *cdka;1* often terminated meiosis shortly after the first meiotic division.

In addition, we observed in the weak loss-of-function *cdka;1* mutants several other defects, e.g., in chromosome condensation. While we cannot exclude that these defects are an indirect consequence of for instance altered ASY1 dynamics, it seems plausible that CDKA;1 has many more roles in meiosis than the here-revealed function in assembling the chromosome axis. Indeed, MLH1 was recently found to be an *in vitro* target of CDKA;1 activity and in *cdka;1* hypomorphic mutants, in which kinase activity is only mildly reduced, an altered recombination pattern with fewer crossovers than in the wild type was observed (Wijnker *et al*, 2019). Interestingly, an alleged increase in CDKA;1 activity also caused an elevation in recombination events hinting at a dosage dependency of CDKA;1 for crossover formation. A key role of Cdks in meiosis is further supported by the large number of meiotic regulators that have Cdk consensus phosphorylation sites and/or a predicted cyclin binding site (Zhu *et al*, 2010). Thus, it seems very likely that we are still at the beginning to understand the phosphorylation control of meiosis by Cdk1-type proteins.

## Materials and Methods

### Plant materials

The *Arabidopsis thaliana* accession Columbia (*Col-0*) was used as wild-type reference throughout this study. The T-DNA insertion lines SALK_046272 (*asy1-4*) (Crismani and Mercier, 2013), SALK_031449 (*pch2-2*) (Lambing *et al*, 2015) and SAIL_423H01 (*asy3-1*) (Ferdous *et al*, 2012), and SALK_106809 (*cdka;1-1*) (Nowack *et al*, 2006) were obtained from the T-DNA mutant collection at the Salk Institute Genomics Analysis Laboratory (SIGnAL, http://signal.salk.edu/cgi-bin/tdnaexpress) via NASC (http://arabidopsis.info/). The mutants *cdka;1 PRO$_{CDKA;1}$: CDKA;1$^{T161D}$* and *cdka;1 PRO$_{CDKA;1}$:CDKA;1$^{T14D;Y15E}$*, the *PRO$_{ZYP1B}$:ZYP1B:GFP* reporter, and the *PRO$_{ASY3}$:ASY3:RFP* reporter plants were described previously (Dissmeyer *et al*, 2007, 2009; Yang *et al*, 2019). The *StrepIII-tag-CDKA;1* (*cdka;1*) line was also generated previously. The StrepIII tag is a Twin-strep-tag® developed by the IBA GmbH, which consists of two tandem Strep II tag moieties separated by a short linker and shows better binding characteristics in comparison with Strep II tag (Pusch *et al*, 2012; Schmidt *et al*, 2013). The protein sequence of StrepIII/Twin-strep-tag is <u>WSHPQFEK</u>-GGGSGGGSGGSA-<u>WSHPQFEK</u> (the Strep II tag moieties are underlined). All plants were grown in growth chambers with a 16-h light/21°C and 8-h/18°C dark cycle at 60% humidity.

## Plasmid construction and plant transformation

To generate the *ASY1* reporters, a 6,013 bp genomic sequence of *ASY1* was amplified by PCR and subsequently integrated into *pENTR2B* vector by SLICE reaction. A *SmaI* restriction site was then introduced in front of the stop codon by PCR. The constructs obtained were then linearized by *SmaI* restriction and ligated with GFP, RFP or mVenus fragments, followed by gateway LR reaction with the destination vector *pGWB501*. The *CDKA;1:mVenus* reporter was generated by using the same strategy as described above. For the *PCH2:GFP* reporter, a 5,837 bp genomic sequence of *PCH2* was amplified by PCR and subsequently integrated into *pDONR221* vector by gateway BP reaction. Subsequently, an *AscI* restriction site was inserted into *pDONR221-PCH2* between the 35–36aa of PCH2 by PCR. Following the linearization by *AscI*, a GFP fragment was inserted into *pDONR221-PCH2*. The resulting *PCH2:GFP* expression cassette was integrated into the destination vector *pGWB501* by the gateway LR reaction. For creating variants of the *ASY1:GFP* constructs including *ASY1^{1–570}:GFP*, a PCR-based mutagenesis was performed using *pENTR2B-ASY1:GFP* as a template followed by gateway LR reactions for integration into the destination vector. All constructs were transformed into *Arabidopsis thaliana* plants by floral dipping.

To make the constructs for the yeast two-hybrid assays, the coding sequences of the respective genes were amplified by PCR with primers flanked by *attB* recombination sites and subcloned into *pDONR223* vector by gateway BP reactions. The resulting constructs were subsequently integrated into the *pGADT7-GW* or *pGBKT7-GW* vectors by gateway LR reactions. Primers used for generating all constructs mentioned above are shown in Appendix Table S1.

## Microscopy and live cell imaging

Light microscopy was performed with an Axiophot microscope (Zeiss). To study protein localization, young anthers harboring the relevant reporters were dissected and imaged immediately using an Leica TCS SP8 inverted confocal microscope. The meiotic stages were determined by combining the criteria of the chromosome morphology, nucleolus position (mainly for pre-meiosis to leptotene), and cell shape. For tracing the dynamics of ASY1:GFP/RFP variants in *asy1* mutants and/or wild-type plants, live cell imaging was performed as described by Prusicki *et al* (2019) under controlled temperature (18–20°C) and humidity (60%) conditions. In brief, one single fresh flower bud was detached from the stem and dissected with two anthers exposed. Subsequently, the isolated bud including the pedicel and a short part of the floral stem was embedded into the *Arabidopsis* apex culture medium (ACM) and then covered by one drop of 2% agarose. The sample was then subjected to constant image capture with 7 min of intervals by using an upright Zeiss LSM880 confocal microscope.

To analyze the distribution of the nucleus versus cytoplasm localized CDKA;1, live cell imaging was performed with two anthers of *cdka;1* mutants harboring a fully functional *CDKA;1:mVenus* reporter for 26 h (Movie EV1). To quantify the subcellular distribution of the CDKA;1:mVenus, the signal intensities in the nucleus and cytoplasm were calculated every hour by segmenting the respective regions using the image processing software Fiji.

## Yeast two-hybrid assay

Yeast two-hybrid assays were performed according to the Matchmaker Gold Yeast two-hybrid system manual (Clontech). Different combinations of constructs were co-transformed into yeast strain AH109 using the polyethylene glycol/lithium acetate method as described in the manual. Yeast cells harboring the relevant constructs were grown on the SD/-Leu-Trp, SD/-Leu-Trp-His, and SD/-Leu-Trp-His-Ade plates to test protein–protein interactions.

## Protein expression and purification

To generate the HisMBP-ASY1, HisMBP-ASY1^{1–300}, HisMBP-ASY1^{1–300/T142V;T184V}, and HisGST-ASY3 constructs, the respective coding sequences were amplified by PCR and subcloned into *pDONR223* vector by gateway BP reactions. The resulting constructs were integrated by gateway LR reactions into *pHMGWA* or *pHGGWA* vectors for the HisMBP- and the HisGST-tagged fusions, respectively. For heterologous expression, the constructs were transformed into the *E. coli* BL21 (DE3) pLysS cells, which were grown at 37°C in the presence of 100 mg/l ampicillin until the $OD_{600}$ of 0.6, followed by protein induction by adding IPTG to a final concentration of 0.3 mM. The cells were further incubated at 37°C for 3 h (HisMBP-ASY1, HisMBP-ASY1^{1–300}, and HisMBP-ASY1^{1–300/T142V;T184V}) or 18°C overnight (HisGST-ASY3). All proteins were purified under native conditions by using Ni-NTA sepharose (QIAGEN) according to the manual.

## GST pull down assays

For GST pull down assay, 4 μg of HisMBP-ASY1^{1–300}, HisMBP-ASY1^{1–300/T142V;T184V}, and 2 μg HisGST-ASY3 were added to the pull down buffer system containing 25 mM Tris–HCl, pH 7.5, 100 mM NaCl, 10% glycerol, and 20 μl GST agarose beads (ChromoTek) as indicated in Fig 4B. After incubation for 1 h at 4°C, GST beads were collected by centrifugation and washed three times with the washing buffer (25 mM Tris–HCl, pH 7.5, 200 mM NaCl, 10% glycerol, and 0.25/0.5/1% Triton X-100). Bead-bound proteins were eluted by boiling in an equal volume of 1X SDS protein loading buffer and subjected to immuno-blotting analysis.

## Protein blots

For SDS–PAGE, protein samples were subjected to the gel electrophoresis at room temperature (12% acrylamide, 375 mM Tris–HCl, pH 8.8, and 0.1% SDS) followed by transfer blotting onto nitrocellulose membrane. For Phos-tag SDS–PAGE, proteins from kinase assays were subjected to Phos-tag gel electrophoresis [6% acrylamide, 375 mM Tris–HCl, pH 8.8, 50 μM Phos-tag (Wako), and 100 μM $MnCl_2$] at 4°C followed by transfer blotting. After incubation with the primary and secondary antibodies, the immuno-blots were exposed and observed using a Bio-Rad Image Analyzer. Relative protein levels were quantified with the Image Lab software (Bio-Rad).

## Chromosome spreads

Chromosome spreads were performed as described previously (Wijnker *et al*, 2012). In brief, fresh flower buds were fixed in 75%

ethanol and 25% acetic acid for 48 h at 4°C, washed two times with 75% ethanol and stored in 75% ethanol at 4°C. For spreading, flower buds were digested in an enzyme solution (10 mM citrate buffer containing 1.5% cellulose, 1.5% pectolyase, and 1.5% cytohelicase) for 3 h at 37°C and then transferred onto a glass slide, followed by mashing with a bended needle. Spreading was performed on a 46°C hotplate by adding 10 μl of 45% acetic acid. The slide was then rinsed with ice-cold ethanol/acetic acid (3:1) solution and mounted in VECTASHIELD with DAPI (Vector Laboratories).

### *In vitro* kinase assays

CDKA;1-SDS, CDKA;1-TAM, and CDKA;1-CYCA3;1 complexes were expressed as described by Harashima and Schnittger (2012). The kinase complexes were purified by Strep-Tactin Agarose (IBA), followed by desalting with PD MiniTrap G-25 (GE Healthcare). The kinase assay in Fig 2B was performed by incubating the kinase complexes with the ASY1 proteins purified from baculovirus-infected insect cells in the kinase buffer containing 50 mM Tris–HCl, pH 7.5, 10 mM $MgCl_2$, and 5% (V/V) [γ-$^{32}$P]ATP (9.25 MBq, GE Healthcare) for 30 min. The reaction was then inactivated by boiling at 95°C for 5 min after adding 5X SDS protein loading solution, and autoradiography was subsequently performed following the SDS–PAGE. The kinase assay in Fig 2C was performed by incubating the kinase complexes with HisMBP-ASY1 in the reaction buffer containing 50 mM Tris–HCl, pH 7.5, 10 mM $MgCl_2$, 0.5 mM ATP, and 5 mM DTT for 90 min. The phosphorylation of ASY1 was then verified by Phos-tag SDS–PAGE. The CBB stained gel after kinase reaction is shown in Appendix Fig S2A.

### Sample preparation and LC-MS/MS data acquisition

The protein mixtures after kinase assays were reduced with dithiothreitol, alkylated with chloroacetamide, and digested with trypsin. Subsequently, the digested samples were desalted using StageTips with C18 Empore disk membranes (3 M; Rappsilber *et al*, 2003), dried in a vacuum evaporator, and dissolved in 2% ACN, 0.1% TFA. Samples were analyzed using an EASY-nLC 1200 (Thermo Fisher) coupled to a Q Exactive Plus mass spectrometer (Thermo Fisher). Peptides were separated on 16 cm frit-less silica emitters (New Objective, 0.75 μm inner diameter), packed in-house with reversed-phase ReproSil-Pur C18 AQ 1.9 μm resin (Dr. Maisch). Peptides were loaded on the column and eluted for 50 min using a segmented linear gradient of 5–95% solvent B (0 min: 5%B; 0–5 min-> 5%; 5–25 min-> 20%; 25–35 min-> 35%; 35–40 min-> 95%; 40–50 min-> 95%; solvent A 0% ACN, 0.1% FA; solvent B 80% ACN, 0.1%FA) at a flow rate of 300 nl/min. Mass spectra were acquired in data-dependent acquisition mode with a TOP15 method. MS spectra were acquired in the Orbitrap analyzer with a mass range of 300–1,500 m/z at a resolution of 70,000 FWHM and a target value of $3 \times 10^6$ ions. Precursors were selected with an isolation window of 1.3 m/z. HCD fragmentation was performed at a normalized collision energy of 25. MS/MS spectra were acquired with a target value of $5 \times 10^5$ ions at a resolution of 17,500 FWHM, a maximum injection time of 120 ms, and a fixed first mass of m/z 100. Peptides with a charge of 1, greater than 6, or with unassigned charge state were excluded from fragmentation for

$MS^2$; dynamic exclusion for 20 s prevented repeated selection of precursors.

For targeted analysis, samples were resolved using the segmented linear gradient as mentioned above. The acquisition method consisted of a full scan method combined with a non-scheduled PRM method. The 16 targeted precursor ions were selected based on the results of DDA peptide search in Skyline. MS spectra were acquired in the Orbitrap analyzer with a mass range of 300–2,000 m/z at a resolution of 70,000 FWHM and a target value of $3 \times 10^6$ ions, followed by MS/MS acquisition for the 16 targeted precursors. Precursors were selected with an isolation window of 2.0 m/z. HCD fragmentation was performed at the normalized collision energy of 27. MS/MS spectra were acquired with a target value of $2 \times 10^5$ ions at a resolution of 17,500 FWHM, a maximum injection time of 120 ms, and a fixed first mass of m/z 100.

### MS data analysis and PRM method development

Raw data from DDA acquisition were processed using MaxQuant software (version 1.5.7.4, http://www.maxquant.org/; Cox and Mann, 2008). MS/MS spectra were searched by the Andromeda search engine against a database containing the respective proteins used for the *in vitro* reaction. Trypsin specificity was required and a maximum of two missed cleavages allowed. Minimal peptide length was set to seven amino acids. Carbamidomethylation of cysteine residues was set as fixed, phosphorylation of serine, threonine and tyrosine, oxidation of methionine, and protein N-terminal acetylation as variable modifications. The match between runs option was disabled. Peptide spectrum matches and proteins were retained if they were below a false discovery rate of 1% in both cases.

The DDA approach only enabled the identification of T142. To analyze the putative phosphorylation sites at T184 and T535, a targeted approach was employed. Raw data from the DDA acquisition were analyzed on MS1 level using Skyline (version 4.1.0.18169, https://skyline.ms; MacLean *et al*, 2010) and a database containing the respective proteins used for the *in vitro* reaction. Trypsin specificity was required and a maximum of two missed cleavages allowed. Minimal peptide length was set to seven maximum length to 25 amino acids. Carbamidomethylation of cysteine, phosphorylation of serine, threonine and tyrosine, oxidation of methionine, and protein N-terminal acetylation were set as modifications. Results were filtered for precursor charges of 2, 3, and 4. For each phosphorylated precursor ion, a respective non-phosphorylated precursor ion was targeted as a control, and several precursor ions from the backbone of ASY1 recombinant protein were chosen as controls between the different samples. In total, 16 precursors were chosen to be targeted with a PRM approach. After acquisition of PRM data, the raw data were again processed using MaxQuant software, with above-mentioned parameters. Peptide search results were analyzed using Skyline using above-mentioned parameters; additionally, data were filtered for b- and y-ions and ion charges +1 and +2.

### Quantification and statistical analysis

Student's *t*-test (two-tailed) was used to evaluate the significance of the difference between the two groups. * denotes $P < 0.05$, and ** denotes $P < 0.01$. The significance of the differences in more than two groups was determined by one-way ANOVA followed by

Turkey's test. Level of significance is indicated by different letters. The numbers of samples are indicated in the figure legends.

## Data availability

The mass spectrometry proteomics data have been deposited to the ProteomeXchange Consortium via the PRIDE partner repository with the dataset identifier PXD011035 (http://proteomecentral.proteome xchange.org/cgi/GetDataset?ID = PXD011035). The results of the mass spectrometry with a targeted approach analyzed using Skyline have been deposited to the Panorama Public (dataset link: https://panoramaweb.org/ASY_phosphorylation.url).

**Expanded View** for this article is available online.

## Acknowledgements

We acknowledge the Salk T-DNA collection, the GABI-Kat T-DNA collection, the *Arabidopsis* Biological Resource Center (ABRC), and the European *Arabidopsis* Stock Centre (NASC) for providing seeds of T-DNA lines used in this report. We thank Dr. Maren Heese (University of Hamburg) and four anonymous reviewers for their comments and constructive feedback on this manuscript. This work was supported by core funding of the University of Hamburg. The IJPB benefits from the support of the LabEx Saclay Plant Sciences-SPS (ANR-10-LABX-0040-SPS).

## Author contributions

CY and AS conceived the experiments. CY, KS, EW, YH, LCa, HH, SCS, DV, LCh, ZO-N, GP, HN, PS, and MG performed the experiments and statistical analyses; SCS and HN performed the mass spectrometry experiment and data analysis. CY, KS, EW, YH, LCa, HH, SCS, DV, LCh, ZO-N, GP, HN, PS, MG, and AS analyzed the data. CY and AS wrote the manuscript.

## Conflict of interest

The authors declare that they have no conflict of interest.

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
