## [Review Process File · The EMBO Journal]

The *Arabidopsis* Cdk1/Cdk2 homolog CDKA;1 controls chromosome axis assembly during plant meiosis

Chao Yang, Kostika Sofroni, Erik Wijnker, Yuki Hamamura, Lena Carstens, Hirofumi Harashima, Sara Christina Stolze, Daniel Vezon, Liudmila Chelysheva, Zsuzsanna Orban-Nemeth, Gaëtan Pochon, Hirofumi Nakagami, Peter Schlögelhofer, Mathilde Grelon and Arp Schnittger.

Review timeline:

Submission date:	24 th January 2019
Editorial Decision:	27 th February 2019
Revision received:	3 rd July 2019
Editorial Decision:	16 th August 2019
Revision received:	2 nd September 2019
Accepted:	4 th September 2019

Editor: Hartmut Vodermaier

Transaction Report:

1st Editorial Decision

27th February 2019

Thank you for submitting your manuscript on CDKA;1 control of meiotic chromosome axis assembly to our editorial office. It has now been evaluated by four referees with expertise in plant cell cycle/cell division and in general meiotic mechanism, whose comments you will find copied below. In light of the generally supportive and constructive feedback from the reviewers, we would be interested in considering a revised version of the manuscript further. However, it is also apparent that all referees bring up a number of substantive concerns regarding both experimental and presentational aspects, which similarly would need to be decisively addressed before publication in The EMBO Journal may be warranted.

I would therefore like to invite you to prepare a revised manuscript along the lines suggested by the reviewers, keeping in mind that it is our policy to allow only a single major revision round and therefore important to diligently and comprehensively answer to all points raised at this stage.

REFeree REPORTS.

Referee #1:

The manuscript "The *Arabidopsis* Cdk1/Cdk2 homolog CDKA;1 controls chromosome axis assembly in meiosis" by Yang...Schnittger presents an interesting series of experiments revealing a role for the *Arabidopsis thaliana* CDK1 protein CDKA;1 in assembly and disassembly of the meiotic chromosome axis. The chromosome axis, and in particular its HORMA domain protein components (ASY1 in *Arabidopsis*) is critical for controlling meiotic crossover formation in most

eukaryotes. In most eukaryotes including plants, the meiotic HORMA domain proteins assemble onto the chromosome axis early in meiosis, then are removed concomitant with assembly of the synaptonemal complex, which links the chromosome axes of paired homologs. HORMA domain proteins are recruited by other axis proteins (ASY3 in Arabidopsis) and also putatively assemble head-to-tail oligomers, and are removed from chromosomes through the action of the AAA+ ATPase Pch2 (TRIP13 in mammals).

Here, Yang et al start with the prior knowledge that mutations in two CDKA;1 cyclins, SDS and TEM, give meiotic defects. They show that CDKA;1 localizes to the nucleus and in particular the axis of unpaired chromosomes in meiotic prophase, and that a weak loss-of-function mutation causes dramatic problems, including in particular a delay in assembly of ASY1 onto meiotic chromosomes. They find that a CDKA;1-SDS complex phosphorylates ASY1 in vitro, and that one site in particular (T142) is important. A T142V mutant of ASY1 does not bind ASY3 in vitro, and shows delayed/defective assembly on meiotic chromosomes. In contrast, a phosphomimic mutant (T142D) binds ASY3 strongly, and assembles on chromosomes properly even in a CDKA;1 mutant. They next present a series of experiments examining the genetic interactions of ASY1 T142 mutants with Pch2, which are somewhat confusing and on the surface appear to contradict some of their earlier findings. Along the way, they demonstrate that the ASY1 HORMA domain binds ASY3 directly, and also binds a short motif at the ASY1 C-terminus (as in related meiotic HORMA domain proteins). Overall, this is an interesting series of results, but I would recommend that the authors complete a few additional experiments before supporting publication.

Major points:

First, a recent paper (West...Corbett eLife 2019) identified the "closure motifs" in both ASY1 and ASY3, and showed that the two motifs are weakly homologous. This should be cited when discussing these data, particularly that in Figure 5.

Page 4: The authors note that "it was recently shown that Hop1 builds a homopolymer through its C-terminal closure motif and it was thought that this polymerization is crucial for its function and axis association since the point mutation K593A in the closure motif of Hop1 causes an 11-fold reduction in CO number and results in high spore lethality". First, the authors should cite Niu...Hollingsworth MBoC 2005 in addition to West et al 2018, as Niu isolated the K593A mutant and showed its genetic effects. Second, West et al (2018) did not formally demonstrate that Hop1 builds a homopolymer: while a Hop1 homopolymer is the most likely explanation for their data, they did not show homopolymer formation directly either in vitro or in cells. Thus, this sentence should be slightly altered.

The authors talk a lot about CDKA;1-mediated phosphorylation of ASY1 as the key mechanism they have discovered. I would recommend shifting the emphasis to talk about phosphorylation of ASY1 by the CDKA;1-SDS complex (this seems to be the key cyclin from their results, but it's not made obvious in the text until perhaps the Discussion section).

Figure 1C - The finding that CDKA;1 seems to localize to the chromosome axis only prior to synapsis is an important one. The figure panel, however, is not very convincing due to low DAPI signal (at least in this reproduction of the figure). The finding would be significantly strengthened if the authors were to also visualize a marker of the synaptonemal complex (e.g. using the authors' ZYP1:GFP reporter)

The observations regarding ASY1 phosphorylation mutants in pch2 mutant strains are interesting, but confusing. First, the authors should note that an earlier study by Lambing...Franklin (PLoS Genetics 2015) showed that Pch2 is required for initial chromosome localization of ASY1 in Arabidopsis. Second, there is a seeming disagreement between the in vitro result that ASY1 T142V weakens ASY3 binding, and the finding that this mutant localizes strongly to chromosomes in a Pch2 mutant. The authors account for this in the Discussion quite nicely, with an interesting idea that Pch2 may mediate low-level axis disassembly early in meiosis unless ASY1 is phosphorylated, and therefore resistant to disassembly. But, ASY1 eventually does come off the chromosomes, indicating either that Pch2 can overcome ASY1 phosphorylation late in meiosis, or that the phosphorylation is lost. In light of this, it would seem that a key missing piece of data is when, exactly, ASY1 is phosphorylated in meiotic prophase? An experiment to answer this question (which would require

phospho-specific antibodies) could indicate whether phosphorylation renders ASY1 more or less amenable to serve as a substrate of Pch2.

Having a phospho-specific antibody would also enable the authors to test whether the weak loss-of-function alleles of CDKA;1 indeed do cause reduced phosphorylation of ASY1 in meiocytes.

ASY1 residue T142 is located in an interesting place on the HORMA domain. Mapping the ASY1 sequence onto the structure of *C. elegans* HIM-3 (using the PHYRE protein structure prediction server) shows that this residue is likely located at the N-terminus of the alpha-C helix, just at the end of the long "beta5-alphaC" loop. Since this loop anchors the C-terminal safety belt in place, one idea might be that phosphorylation of T142 imparts greater flexibility on the loop, and thereby may allow the safety belt to disengage to "open" the protein and allow closure motif binding/dissociation.

Minor points:

It would probably improve readability if the authors referred to the CDKA;1 T161D mutant as "CDKA;1-D" or even CDKA;1-T161D instead of just "D"

In the introduction, it's not necessary to describe the full origin of the names of each gene (e.g. "Nijmegen Breakage Syndrome 1/X-ray sensitive 2") - this is a bit distracting. Including full names for Arabidopsis genes is fine.

Page 7, top: the authors refer to a previously published StrepIII-tag-CDKA;1 fusion construct. This is indeed what is mentioned in the referenced paper (Pusch 2012), but there's no mention of a "Strep III" tag in any other publication by that group or any others (or indeed, anywhere else on the internet). There is a well-established Strep II tag, which is perhaps what Pusch et al. meant? Please verify what exactly this tag is.

There are several misspellings that would be caught by a spellchecker. For example:

Page 10, the word "restauration" should be "restoration"

Page 16, "essention" should be "essential"

Other typos:

Page 17, first line - "cohesion" should be "cohesin"

Page 17, second paragraph - "evaded" should probably be "evicted"

Referee #2:

The manuscript entitled "The Arabidopsis Cdk1/Cdk2 homolog CDKA;1 controls chromosome axis assembly in meiosis", written by Yang et al., describes the role of CDKA;1 in phosphorylation and molecular dynamics of ASY1, the axial component of meiotic chromosomes in Arabidopsis. At first, to delimit the meiotic stage where CDKA;1 interacts with meiotic chromosomes, the authors introduced CDKA;1 variants with less kinase activity into *cdka;1* null mutants, and the hypomorphic mutant phenotypes and subcellular CDKA;1 localization suggested that CDKA;1 physically interacts with the chromosome axis during early meiotic prophase. Next, they examined the role of CDKA;1-dependent phosphorylation of ASY1 on meiotic chromosome-axis assembly, by introducing the fluorescent ASY1 fusion proteins with various mutations in ASY1 phosphorylation sites, and concluded that the phosphorylation site in the HORMA domain, T142, is major for phospho-regulation of ASY1, with the site T184 having an ancillary role.

In addition to the CDKA;1 role, the authors asked the role of ASY3 and PCH2 in ASY1 dynamics in axis assembly. From careful observations of localization patterns of ASY1 variants in the *pch2* mutant, they concluded that PCH2 plays dual roles and Y2H analyses; one is in helping ASY1 to accumulate into the nucleus, and another is in promoting the release of non-phospho ASY1 from the axis, and simultaneously, increasing the binding affinity of phospho ASY1 to ASY3. Furthermore, they proved the importance of the C-terminal closure motif in self-polymerization and chromosome association of ASY1.

All experiments were elaborately designed and performed, and the results were enough for supporting the conclusions. The information in this paper will be beneficial for all readers studying meiosis not only in plants but also in non-plant species. I wrote several minor comments below;

- (1) In Fig 1C pictures, it was difficult for me to confirm the localization of linear structure of CDKA;1 on DAPI-stained chromosomal threads. I think it better to enlarge several regions and make co-localization clearer, or to show co-localization with some axial components, such as ASY1 and ASY3.
- (2) L20 and 21, p. 14: ASY 1-570  ASY1 1-570.
- (3) L25, p. 16: essention  essential?

Referee #3:

The Arabidopsis Cdk1/Cdk2 homolog CDKA;1 controls chromosome axis assembly in meiosis
 Chao Yang, Kostika Sofroni, Erik Wijnker, Yuki Hamamura, Lena Carstens, Hirofumi Harashima, Sara Christina Stolze, Daniel Vezon, Liudmila Chelysheva, Zsuzsanna Orban-Nemeth, Gaëtan Pochon, Hirofumi Nakagami, Peter Schlögelhofer, Mathilde Grelon and Arp Schnittger

It is now well established that the proteins that comprise the meiotic chromosome axis not only play an important role in the organization of the chromosomes during prophase I but are also central to the controlled formation of genetic crossovers. In many organisms including Arabidopsis, the central components of the chromosome axis have been identified however many aspects of their functionality remain to be resolved. In previous studies the Schnittger group had demonstrated a role for the Cdk1/Cdk2 homolog, CDKA;1, during meiosis in Arabidopsis thaliana. In this manuscript Yang et al have further investigated the role of the kinase in relation to its interplay with the meiotic chromosome axis. As a result the work brings new insight into the regulation of the assembly of the axis. The work reveals that the HORMAD protein ASY1 is at least in part regulated by CDKA;1 phosphorylation and that this is important for the formation of the chromosome axis.

Specific comments

CDKA;1 is dynamically localized during meiosis:

- i. The authors report a change in the comparative intensity of the CDKA;1:mVenus signal in the nucleus relative to the cytoplasm during prophase I. Based on the intensity levels it would seem that the maximum differential is around 1.5X. Some comment as to why and how the authors think this is of functional significance should be included.
- ii. In Fig. 1c the authors use immunolocalization to show that CDKA;1 appears associated with the chromosome unsynapsed chromosome axis. Some regions appear to be devoid of signal and they state that these correspond to the synapsed regions. They rely on the thickness of the DAPI staining to identify the synapsed regions. This may be the case, but to be certain they should use a synaptonemal complex specific marker, such as ZYP1 to confirm this point.

Meiosis is severely affected in hypomorphic *cdka;1* mutants:

- i. The authors report that loss of CDKA;1 activity disrupts meiosis with a loss of crossover formation. Based on analysis of DAPI-stained chromosome spreads they report an absence of synapsis at the "zygotene-like stage". Inspection of panel 1Dh does however show regions of aligned chromosomes that have an intensity of staining not unlike the wild-type control, albeit much less complete. Moreover the authors then go on to say that at pachytene the chromosomes are "largely unpaired". This appears to contradict the previous statement. It does seem there is a synaptic defect however further characterization using a ZYP1 marker would clarify just how much, if any, synapsis is occurring. They should also stick to using the word synapsed rather interchangeably using synapsed or paired. The latter could refer to aligned chromosomes that are pre-synaptic or asynaptic.

Minor point: figure 1D requires a scale bar.

- ii. The authors then go on to say that based on immunolocalization of MRE11 and DMC1 (again no scale bar) see Fig S1e, that these proteins are correctly localized suggesting DSBs are normal. However, they do not provide a wild-type control and there are no counts of foci. The figure indicates that there are foci on the chromatin but in the absence of an axis marker they cannot confirm that the foci are correctly localized. Importantly, they appear to have overlooked the previous observation by Lohmiller et al (Chromosoma 2008, 117; 277-288) that MRE11 forms foci in the absence of DSBs, hence it is not the best choice for demonstrating DSB formation. Ideally, this needs to be carried out using anti-gamma H2AX with proper counts. Similarly the double

mutant with rad51, (no scale bars) does appear to show fragments but whether or not this is the same as the rad51 single mutant cannot be ascertained in the absence of a control and counts. While I would agree that the data suggest DSBs are formed in the *cdka;1* mutant, the data provided is insufficient to conclude that breaks form as normal.

Phosphorylation of ASY1 by CDKA;1 is essential for its recruitment to the chromosome axis

i. The authors write that based on the mutant phenotype and the presence of a CDC28 phospho site in the yeast Hormad protein, Hop1 that this led them to identify ASY1 as a potential target for CDKA;1. They should also mention that a previous study of *in vivo* phosphorylation of ASY1 (Osman et al. *Plant Journal* 2016, 93, 17-33) had identified the protein as a potential CDK1 (and ATM/ATR) target. Although this work was based on the closely related species *Brassica oleracea*, the site in question, T535 is conserved in *Arabidopsis*. Of course, as it turns out, it does appear that based on this study T535 doesn't seem to be crucial for ASY1 function. That the other sites CDKA;1 sites identified here were not identified in the earlier study may simply be due incomplete coverage of the ASY1 in the MS analysis or perhaps indicate that the CDKA;1 phosphorylation of ASY1 is relatively transient. Also it is worth pointing out the Osman study identified CDKA;1 as a putative ASY1 interacting protein.

ii. The authors commence by investigating the localization of a tagged ASY1 in the weak *cdka;1* mutant compared to wild-type. They report a delay in localization based on the extension of the axis associated ASY1 signal relative to the cell morphology and migration of the nucleolus to the side of the nucleus. Previous studies have determined the timing (in hours) of meiotic events based on labeling the meiocytes during S-phase with a marker such as BrdU or EdU rather than the approach used here. Can the authors be sure that the phenotypic events used to judge meiotic progression occur in the wild-type and *cdka;1* with the same timing? If so, do they have evidence to support this?

iii. The authors identify 5 putative CDKA;1 sites in ASY1 and provide evidence that two of these are phosphorylated *in vitro* by the kinase. Importantly, one of these T142, turns out to be the regulatory site. The other site corresponds to that identified in the *in vivo* analysis by Osman et al. Curiously, T184 which they later show to enhance the effect the of T142V mutation was not found to be phosphorylated by CDKA;1 *in vitro*. Do the authors have an explanation for this? CDKA;1

iv. Functional analysis of mutant lines in which combinations of the phosphosites have been mutated provides good support for the key role of T142/T184. As well as impacts on fertility, the authors point to reduced ASY1 polymerization plus more nucleoplasmic signal (Fig 3B) and partial synapsis (Fig S6). With regard to figure 3B how was the reduction in linear ASY1 quantified? Ideally measurements are required particularly as the quality of images obtained by confocal microscopy lack clarity making it difficult for the reader. Also the use of ASY3 as a marker for synapsis is not ideal as it is not possible to fully follow synapsis with this marker particularly the zygotene to pachytene transition. The use of either their SC marker ZYP1-GFP construct (Fig S3a) or an anti-ZYP1 Ab is far more definitive for staging this progress. The same comment applies to Fig S6, the use of DAPI-stained spreads to monitor the degree of synapsis is less than ideal.

Phosphorylation of ASY1 increases its binding affinity with ASY3

i. The authors use Y2H to investigate the interaction of ASY1 and ASY3. Their data indicate that the T142V mutation reduces the interaction and this is restored in the T142S and T142D versions. Does this imply that residue T142 is also phosphorylated in budding yeast? Given that budding yeast can phosphorylate some proteins from higher eukaryotes on T/S residues it does seem reasonable to infer that this is the case here. However, to confirm this is the case the authors could provide evidence of the phosphorylation status of ASY1 in the yeast host.

Phosphorylation of ASY1 counteracts the action of PCH2 in early but not late prophase

i. The authors make the novel and interesting observation that the lack of chromosome localization of the ASY1-T142V/T184V construct observed in wild type and *asyl1* mutant plants is not found in the absence of PCH2 and that it loads on to the chromosomes as efficiently as wild-type ASY1. They then go on to demonstrate using an ASY1-GFP construct that the tagged protein is found in both the nucleus and cytoplasm in the *pch2* mutant. They report a similar accumulation of ASY1-T142V/T184V in the *pch2* mutant. Based on this they suggest that PCH2 facilitates nuclear localization of ASY1. Given the observation that chromosomal localization of ASY1-T142V/T184V in the *pch2* mutant appears normal, does this suggest that PCH2 is not a major player in this respect of ASY1 nuclear localization or its role is non-essential? Or maybe its role is to facilitate ASY1 turnover in the cytoplasm rather than nuclear localization?

Self-assembly of ASY1 through its C-terminal closure motif is affected by the phosphorylation in the HORMA domain.

This section deals with the control of axis assembly and the effect of phosphorylation on the ASY1 closure motif. Initially the authors define a region in the C-terminal of ASY1 that contains a potential closure motif as previously described in some other Horma proteins. However, they do not mention the published work by the Corbett group (BioArchive 2018/Elife 2019) which has previously defined the ASY1 closure motif. This should be corrected. Nevertheless, they do go on to provide some interesting new insights into the role of the closure motif in relation to PCH2 activity. Furthermore they provide evidence to suggest that phosphorylation of the sites in the Horma region of ASY1 is important for interaction with the closure domain.

In summary the authors provide good evidence to indicate an important regulatory role for CDKA;1 in the regulation of ASY1 during axis assembly. As they conclude in their discussion it does seem that the role of the kinase is in the assembly of ASY1 in the axis. It appears not to be important for removal of ASY1 at the synaptic fork at the zygotene/pachytene transition. Whether it is important for maintenance of ASY1 in the axis prior to synapsis remains to be seen. They raise the possibility that the removal of ASY1 at zygotene may require other post-translational modifications. This notion is consistent with the studies by Osman et al who identified a number of other phosphosites within ASY1. This could be mentioned.

As detailed above there are a few issues with the manuscript in its present form. Nevertheless, once these have been addressed, this is a valuable contribution to our understanding of how the formation of the meiotic chromosome axis is controlled in higher plants.

Minor points

- i. There are a number of typographical errors in the text.
- ii. The absence of error bars on some figures needs to be corrected.
- iii. Details of the ZYP1B-GFP construct are not provided

Referee #4:

The manuscript by Yang et al uses genetic, cytological and biochemical approaches to analyze the role that the cyclin-dependent kinase, CDKA;1 plays in chromosome synapsis in *Arabidopsis thaliana*. They show that CKDA;1 localizes to meiotic nuclei at the same time as the HORMA domain containing axial element protein, ASY1. Furthermore, hypomorphic alleles of CKDA;1 exhibit synapsis defects that are downstream of DSB and resection. Phenotypic analysis of the five putative Cdk sites in ASY1 revealed a role for T132 in the HORMA domain in chromosome synapsis, with a contributing effect for T184. Non-phosphorylatable alleles at these sites were generated by substituting valine for threonine and the double mutant showed defects in chromosomal localization *in vivo*, as well as interaction with a second axial element protein, Red1 in genetic and biochemical assays. Negative charges at T132 and T184 resembled wild-type and even promoted ASY1-ASY3 binding in pulldown assays. Interestingly the chromosome localization defect of the *asy1-T132V T184V* mutant was rescued by mutation of the PCH2 gene. Finally, this paper shows that the HORMA domain of ASY interacts with a C-terminal fragment of ASY1, which the authors propose is a closure motif. These results are interesting and if the interpretations of the data are accurate would be a strong contribution to our understanding of meiotic chromosome behavior in plants. However, there are several concerns that need to be addressed. Most importantly, the authors need to show that it is the inability to be phosphorylated, and not just the valines, that are responsible for the defects (see below). In addition, the scholarship of the manuscript is poor with many citations missing, some that are irrelevant, and some that do not support the facts for which they are being cited.

Major comments:

1. The conclusion that CDKA;1 is a "master regulator of meiosis" is too strong given that what is shown is that it regulates protein-protein interactions between ASY1 and itself and ASY3. While this is an important aspect of meiotic prophase, it is only one part of a highly regulated process. A

master regulator should be at the top of the hierarchy that controls everything. For example, while Mer2 is constitutively phosphorylated by Cdk in yeast, its role is to provide the priming phosphorylation for the Cdc7-Dbf4 kinase that travels with the replication fork. In this situation, it is really DDK which is the regulator providing the readout for replication.

2. Beginning of last paragraph on page 4: The authors are confusing the central element with the central region. The central element is a dark staining linear structure located midway between, and running parallel to, the lateral elements. The central element is comprised of different proteins from the central region. The central region is created by the insertion of perpendicular transverse filament proteins such as Zip1/ZYP1. It is the transverse filament proteins of the central region that hold the lateral elements together. Since many Zip1 dimers polymerize to make the central region, it would be more accurate to say the central region is formed by Zip1 oligomers.

3. At the end of the discussion the authors state that they show that CDKA;1 is "especially important for chromosome pairing and synapsis". Pairing is functionally distinct from synapsis and is the early alignment of homologous chromosomes prior to synapsis. To determine whether pairing has occurred, FISH experiments are necessary to show that homologous sequences have aligned. These types of experiments were not done. I agree with the authors' conclusions that negative charges on ASY1 promote synapsis, but see no evidence to support the claim about pairing.

4. Figure 1A and movie: The authors claim that CDKA;1 is "dynamically localized during meiosis" and that "in early prophase, CDCKA;1mVenus is equally distributed between the nucleus and the cytoplasm". I don't see this and don't understand the diagrams trying to explain it. The cartoon for early prophase in Figure 1A shows a beige rectangle with a green oval that contains within it a white oval. These different parts are not labeled in the figure legend. My assumption is the beige rectangle is the cytoplasm and the white circle is the nucleolus. It is clear in both the movie and Figure 1 A that there is more green color in the nucleus compared to either the cytoplasm or the nucleolus. I see some changes in abundance, but this is not the same as "dynamic localization" which would suggest the kinase is moving from one location to another.

5. In Figure 1C, the authors claim that CDKA;1 is removed from areas of the chromosomes that have synapsed. The basis for determining what is synapsed and what is not is not clear. This is an important conclusion since it correlates with the localization of ASY1 and is part of their argument that CDKA;1 regulates ASY1 localization. Therefore the relationship of CDKA;1 localization to chromosome synapsis should be determined by co-localization experiments with ZYP1 to show that the two do not co-localize (similar to the experiment that the authors did with ASY1 and ZYP1).

6. Figure 1D shows a striking phenotype for the *cdka;1-D* mutant where there are 10 univalents instead of 5 bivalents (panels k vs d). After this statement it says (3%; n = 7). What does this mean? Did only 3% of cells exhibit this phenotype? What did the other 97% look like? Does n= 7 mean that 7 spreads were examined? If so, where does the 3% come from?

7. It is not clear what is going on with the *in vitro* kinase assay in Figure 2C. According to the Figure legend, the top panel is an autoradiograph, suggesting that phosphorylation is being monitored by the transfer of a radioactive phosphate onto recombinant ASY1. However, in the methods section there is no mention of the use of radioactive ATP. Assuming I have interpreted the assay correctly, then there should be no signal for ASY1 in the autoradiograph in the absence of a CDK. However there is clear ASY1 band in Lane 1 and also in Lane 4 which the authors claim is a negative control. Further explanation is necessary for the reader to understand this experiment.

8. Figure 4 legend: The title overinterprets the data and should be changed to something like "A negative charge at T132 in the HORMA domain of ASY1 promotes interaction with ASY3". The experiments in this panel use aspartic acid to mimic phosphorylation instead of assaying phosphorylation directly. Binding affinity is a specific parameter which needs to be measured biochemically. The quantification of the binding experiments in 4B is not clear. How was the ASY1 normalized? It should be to the amount of Red1 in the pulldown. 4C graphs "Relative levels of ASY1"-relative to what? What Triton-X 100 concentration was shown for the graph?

9. An important question is whether the interpretation that defects observed for the ASY1 mutants with valine substitutions for CDK phosphosites is due to a lack of phosphorylation is correct. A

recent paper (Winters and Pryciak, Mol Biol Cell 2019) showed that valine mutations in STE5 resulted in hyperactive signalling in the yeast pheromone response pathway but that this was not observed when the putative phosphosites were replaced with either alanine or glycine. The idea is that valine has a more hydrophobic side chain than serine or glycine and that this can affect protein function beyond not getting phosphorylated. This (hopefully unlikely) possibility can be tested by mutating T132 and T184 in the ASY1 2-hybrid construct and showing that this also abolishes the interaction with ASY3. If this is not the case, then the major conclusions of the paper are in doubt.

10. Does the C-terminal ASY1 fragment proposed to contain the closure motif have sequence similarity with the closure motif consensus published by West et al? If so, that would make the authors' case more compelling.

Minor comments:

Page 3, The Roeder lab paper describing Mer2 should be cited after the statement that it is a substrate for Cdk.

References for the first sentence in paragraph 3 about Cdk phosphorylating Sae2 are needed.

References are needed for the statement that "single stranded ends are captured by recombinases Dmc1 and Rad51....".

Line 31, Heteroduplex DNA is not "resolved", Holliday junctions are.

Page 4, top: The statement that "reducing Cdc28 activity...did not affect the number of Rad51 foci" contradicts the earlier assertion that Cdk activity is required to make DSBs. Also please indicate in which organisms these results were generated.

The authors should define Cdc28 as the catalytic subunit of Cdk in yeast. Then throughout, for example top of page 5, it would be more accurate to say phosphorylation by Cdk, not Cdc28, since Cdc28 is not active on its own.

Line 13: The authors should cite the Rockmill and Roeder 1990 Genetics paper, and the Smith and Roeder 1997 JCB paper which showed that Red1 is part of the axial core and not on chromatin. Hollingsworth and Johnson is not relevant here.

Line 15: The authors should cite the Hollingsworth et al 1990 Cell paper for Hop1 in yeast.

Line 20: Smith and Roeder 1997 should be cited for the requirement of RED1 for Hop1 to bind to chromosomes. Again, Hollingsworth and Johnson is not relevant here.

End of paragraph: Niu et al 2005 should be cited for the meiotic phenotypes of the hop1-K593A mutant.

The authors should cite the Toth 2009 PLoS Genetics paper for the loss of the HORMAD proteins from synapsed chromosomes which is dependent upon the Pch2 ortholog.

Page 5, first paragraph and throughout: instead of "de-phospho" which is non-standard usage, say "unphosphorylatable".

Line 9:, ...several Cdks...

A style suggestion to make the writing more concise-remove words like "has been found, ...were found...etc". So the second paragraph on page 5 could read: The model plant...and some of them function in meiosis. Six out of the ten...are expressed in meiosis.... However of these eight...and SDS exhibit meiotic defects."

Page 6, top: need refs for the relationship of CDKG to human Cdk10 and for reason why it is not part of the core cell cycle machinery.

Page 7, line 6, "After synapsis of the homologs".

Last paragraph, please indicate some of the mitochondria shown in Figure 1D with arrows.

Page 8: first paragraph, "The absence of synapsis and...."

Line 26, saying that CDKA;1 is "indispensable...for chromosome synapsis and bivalent formation" is too strong given that it appears that many cells in the mutant are capable of these processes

Line 30, again, pairing was not assayed and therefore cannot be claimed to be affected.

Page 9, line 5: needs a citation for the fact that asy1 mutants are asynaptic

Line 11, In yeast, Hop1 and Red1 were shown to localize to the axis, not to chromatin (see the Smith and Roeder 1997 paper). The chromatin is in the loops that emanate out of from the axis. Do the authors think that ASY1 binds to chromatin as well as the axis?

Page 10, top "Successful" not "Succeeding"

Line 3, remove "strongly" phosphorylated since no quantification is presented. Also see Major comment 7 about my confusion with the interpretation of this experiment.

Line 19: ...was unaffected in the plants expressing...

Line 20: not clear what is meant by the verb "collate" in this context

Line 24: restoration, not restauration

Line 25: silique not siliques

Line 27: ...during leptotene similarly to ASY:GFP...

Line 32: Cite figure for failure of the 5V mutant to localize

Page 11, line 4, wild-type

Line 21, ...was no longer phosphorylated...

Page 12, line 2: delete "strongly"

Line 7, it would be helpful if the authors explained the details of their two hybrid assay. It appears there are two reporters and that the single -His selection is less stringent than the double selection for histidine and adenine prototrophy, which is why they conclude that the T142V T184V mutant has a more severe interaction defect than T142 alone.

Figure 4A: It is interesting that the T142 and S142 ASY1 truncations interact with ASY3 in the two-hybrid system, since this suggests that these amino acids are phosphorylated in mitotic yeast cells (unless it is because they are not valines). Therefore the specificity of the cyclin does not appear to be critical for this modification.

Line 26, phospho-mimicking, not phospho-mimicry

Page 13, line 13, necessity

Page 14, top, Need REFs for chromosomal localization of Hop1 and the HORMADs in yeast (Smith and Roeder, 1997 and Toth paper).

Bottom, To my eye, the growth of the T142D mutant is not as robust as the wild-type, so saying the two showed "similar interaction strength" is not consistent with the data.

Page 16, line 27, essential, not essention

Page 17, line 4: the authors say: "...binding capacity...was enhanced when the closure motif of ASY1 was deleted". This is an over-interpretation of the data. Binding capacity was enhanced when a terminal region of ASY1 was deleted. The authors hypothesize that this is because the terminal domain contains a closure motif but this has not been proven. If this region contains the sequence determinants previously shown to be present in a closure motif, then this conclusion would be warranted.

Line 9, What does "at least in yeast" mean in this sentence which is talking about ASY1 interaction?

Second paragraph, It is not true that in yeast PCH2 is required for synapsis and recombination. Several papers have shown that pch2 mutants in yeast exhibit more crossovers than wild-type (one of them being the Joshi et al reference that is cited). Synapsis also occurs, but instead of alternating Hop1 and Zip1 domains, Zip1 staining is continuous.

Line 20, not sure what "evaded" means in the context of this sentence

Page 18, line 2 not sure what "partition" means in the context of the sentence

Bibliography, gene and genus species names should be italicized. Only proper nouns and the first word of the title should be capitalized.

Detailed response to the reviewer comments on Yang et al.**Referee #1:**

The manuscript "The Arabidopsis Cdk1/Cdk2 homolog CDKA;1 controls chromosome axis assembly in meiosis" by Yang...Schnittger presents an interesting series of experiments revealing a role for the Arabidopsis thaliana CDK1 protein CDKA;1 in assembly and disassembly of the meiotic chromosome axis. The chromosome axis, and in particular its HORMA domain protein components (ASY1 in Arabidopsis) is critical for controlling meiotic crossover formation in most eukaryotes. In most eukaryotes including plants, the meiotic HORMA domain proteins assemble onto the chromosome axis early in meiosis, then are removed concomitant with assembly of the synaptonemal complex, which links the chromosome axes of paired homologs. HORMA domain proteins are recruited by other axis proteins (ASY3 in Arabidopsis) and also putatively assemble head-to-tail oligomers, and are removed from chromosomes through the action of the AAA+ ATPase Pch2 (TRIP13 in mammals).

Here, Yang et al start with the prior knowledge that mutations in two CDKA;1 cyclins, SDS and TEM, give meiotic defects. They show that CDKA;1 localizes to the nucleus and in particular the axis of unpaired chromosomes in meiotic prophase, and that a weak loss-of-function mutation causes dramatic problems, including in particular a delay in assembly of ASY1 onto meiotic chromosomes. They find that a CDKA;1-SDS complex phosphorylates ASY1 in vitro, and that one site in particular (T142) is important. A T142V mutant of ASY1 does not bind ASY3 in vitro, and shows delayed/defective assembly on meiotic chromosomes. In contrast, a phosphomimic mutant (T142D) binds ASY3 strongly, and assembles on chromosomes properly even in a CDKA;1 mutant. They next present a series of experiments examining the genetic interactions of ASY1 T142 mutants with Pch2, which are somewhat confusing and on the surface appear to contradict some of their earlier findings. Along the way, they

demonstrate that the ASY1 HORMA domain binds ASY3 directly, and also binds a short motif at the ASY1 C-terminus (as in related meiotic HORMA domain proteins). Overall, this is an interesting series of results, but I would recommend that the authors complete a few additional experiments before supporting publication.

We are very happy for the positive feedback and like to thank this reviewer for his/her constructive comments.

Major points:

First, a recent paper (West...Corbett eLife 2019) identified the "closure motifs" in both ASY1 and ASY3, and showed that the two motifs are weakly homologous. This should be cited when discussing these data, particularly that in Figure 5.

We are aware of this paper, which was deposited in bioRxiv at the moment of the submission of our work and has now been published in eLife. We cite this work in our revised manuscript.

Page 4: The authors note that "it was recently shown that Hop1 builds a homopolymer through its C-terminal closure motif and it was thought that this polymerization is crucial for its function and axis association since the point mutation K593A in the closure motif of Hop1 causes an 11-fold reduction in CO number and results in high spore lethality". First, the authors should cite Niu...Hollingsworth MBoC 2005 in addition to West et al 2018, as Niu isolated the K593A mutant and showed its genetic effects.

We thank the reviewer for highlighting the work by Niu et al. and we cite this paper now.

Second, West et al (2018) did not formally demonstrate that Hop1 builds a homopolymer: while a Hop1 homopolymer is the most likely explanation for their data, they did not show homopolymer formation directly either in vitro or in cells. Thus, this sentence should be slightly altered.

We have rephrased this sentence and write now: ... it was recently proposed that Hop1 might build a homopolymer through its C-terminal closure motif. This polymerization is thought to be crucial for Hop1 function and axis association since the point mutation K593A in the closure motif of Hop1 causes an 11-fold reduction in CO number and results in high spore lethality (Niu et al., 2015; West, et al., 2018)...

The authors talk a lot about CDKA;1-mediated phosphorylation of ASY1 as the key mechanism they have discovered. I would recommend shifting the emphasis to talk about phosphorylation of ASY1 by the CDKA;1-SDS complex (this seems to be the key cyclin from their results, but it's not made obvious in the text until perhaps the Discussion section).

While we show that ASY1 is phosphorylated by a CDKA;1-SDS complex, it is very likely that CDKA;1 also works together with other cyclins to phosphorylate ASY1. Moreover, the *sds* mutant phenotype is not a subset of the phenotype of the weak loss-of-function *cdka;1* mutants as seen by the apparently correct localization of DMC1 in *cdka;1* and the DMC1 localization defects in *sds* mutants (DeMuyt et al., 2009). Conversely, no obvious alteration in ASY1 localization was found in *sds* single mutant indicating that ASY1 can be phosphorylated by other CDKA;1-cyclin complexes in the absence of SDS. Therefore, we hope that the reviewer agrees that it is more accurate to focus on the CDKA;1 part.

Figure 1C - The finding that CDKA;1 seems to localize to the chromosome axis only prior to synapsis is an important one. The figure panel, however, is not very convincing due to low DAPI signal (at least in this reproduction

of the figure). The finding would be significantly strengthened if the authors were to also visualize a marker of the synaptonemal complex (e.g. using the authors' ZYP1:GFP reporter)

We detected CDKA;1 with an anti-Strep antibody using material fixed in Carnoy's fixative (EtOH:Ac = 3:1). Unfortunately, the anti-ZYP1 antibody does not work (or very irregularly) in our hands with this acetic acid spreading technique. To still address this important point of this reviewer, we performed co-immunolocalization of ASY1 and CDKA;1 building on the well-known observations in different species including yeast, mammals and plants that ASY1 and its orthologues are largely depleted from the synaptic regions concomitant with the polymerization of ZYP1 indicating the synapsis. Our co-immunolocalization experiments, presented in Figure 1C of the revised manuscript, show that CDKA;1 co-localizes with ASY1 on the chromosome axis at leptotene, and is largely absent from the synaptic regions, highlighted by the removal of ASY1, at zygotene. We think that these localization experiments substantiate our initial observation that CDKA;1 is removed from chromosomes at the moment of synapsis.

The observations regarding ASY1 phosphorylation mutants in *pch2* mutant strains are interesting, but confusing. First, the authors should note that an earlier study by Lambing...Franklin (PLoS Genetics 2015) showed that Pch2 is required for initial chromosome localization of ASY1 in Arabidopsis.

We have double-checked the publication from Lambing et al. and to our understanding they conclude that the localization/assembly of ASY1 in *pch2* at leptotene is indistinguishable from that in the wildtype.

Second, there is a seeming disagreement between the in vitro result that ASY1 T142V weakens ASY3 binding, and the finding that this mutant localizes strongly to chromosomes in a Pch2 mutant. The authors account

for this in the Discussion quite nicely, with an interesting idea that Pch2 may mediate low-level axis disassembly early in meiosis unless ASY1 is phosphorylated, and therefore resistant to disassembly.

The observation that the compromised chromosome association of ASY1-T142V T184V in the wildtype is restored in the absence of PCH2 suggests that the defective localization of ASY1-T142V T184V depends on PCH2, implying that PCH2 functions also to remove a portion of ASY1 in early meiosis, very likely the non-phosphorylated one. This possibility is also consistent with the presence of PCH2 already in early meiosis (Figure 5D). The rescue of the ASY1-T142V T184V localization by the depletion of PCH2 is further corroborated by the restoration of the fertility of *asy1* mutants harboring ASY1-T142V T184V:GFP when PCH2 is co-depleted (Figure 5B). This also indicates that ASY1-T142V T184V:GFP is largely functional as long as it localizes properly on the chromosome axis. Perhaps some of the confusion in the initial submission arose due to the observation of the compromised binding of the phospho-mutant of ASY1 to ASY3. However, this binding is only, yet clearly, reduced and not abolished. We thank the reviewer for pointing out that we were not clear enough in our description and have tried to describe the results of the Y2H interaction assay in more detail.

But, ASY1 eventually does come off the chromosomes, indicating either that Pch2 can overcome ASY1 phosphorylation late in meiosis, or that the phosphorylation is lost. In light of this, it would seem that a key missing piece of data is when, exactly, ASY1 is phosphorylated in meiotic prophase? An experiment to answer this question (which would require phospho-specific antibodies) could indicate whether phosphorylation renders ASY1 more or less amenable to serve as a substrate of Pch2. Having a phospho-specific antibody would also enable the authors to test whether the weak loss-of-function alleles of *CDKA;1* indeed do cause reduced phosphorylation of ASY1 in meiocytes.

The observation that the phosphorylation-mimicking ASY1-T142D could be removed from the chromosomes at zygotene onwards as the wild-type version, suggests that PCH2 can likely overcome the stronger binding of the phosphorylated ASY1 towards ASY3 at the synaptic regions. This suggests either that an unknown regulator/cofactor of PCH2 exists, which enhances the activity of PCH2, or that PCH2 has a higher activity at the synaptic regions. The latter is supported by the observation that PCH2 specifically accumulates at the synaptic regions at zygotene coinciding with the removal of ASY1 (Figure 5D).

While we agree that a phospho-specific antibody would be very helpful to elucidate the phospho-dynamics of ASY1 during prophase I, we unfortunately don't have such an antibody in our hands nor, at least to our knowledge, does it exist in another lab at the moment. We hope that the reviewer agrees that the generation of such an antibody is a lengthy endeavor with no guarantee for success. Notably, peptide antibodies against phosphorylated amino acid residues are likely to produce background when used in immuno-localization studies. Especially in our case, such an antibody must be extremely well working to detect quantitative differences in phosphorylation levels. We further think that the different lines of experiments we present provide sufficient evidence for a CDK-dependent phospho-control of ASY1. We hope that the reviewer agrees that a detailed analysis of how ASY1 is removed from the axis opens a new chapter that can be studied in the future.

ASY1 residue T142 is located in an interesting place on the HORMA domain. Mapping the ASY1 sequence onto the structure of *C. elegans* HIM-3 (using the PHYRE protein structure prediction server) shows that this residue is likely located at the N-terminus of the alpha-C helix, just at the end of the long "beta5-alphaC" loop. Since this loop anchors the C-terminal safety belt in place, one idea might be that phosphorylation of T142 imparts greater flexibility on the loop, and thereby may allow the safety belt to disengage to "open" the protein and allow closure motif binding/dissociation.

We very much thank the reviewer for this very interesting comment and would like, with the permission of this reviewer, to include this consideration in our discussion with acknowledgements to this and the other reviewers.

Minor points:

It would probably improve readability if the authors referred to the CDKA;1 T161D mutant as "CDKA;1-D" or even CDKA;1-T161D instead of just "D"

We refer now to the *cdka;1* hypomorph as *CDKA;1^{T161D}*.

In the introduction, it's not necessary to describe the full origin of the names of each gene (e.g. "Nijmegen Breakage Syndrome 1/X-ray sensitive 2") - this is a bit distracting. Including full names for Arabidopsis genes is fine.

We have double-checked all gene names but are not entirely sure how to address this comment. Hence, we would like to leave the way of how gene names are introduced to the editor. In any case, it feels a bit awkward to make a difference between gene names first or exclusively found in plants versus gene names stemming from yeast or animals.

Page 7, top: the authors refer to a previously published StrepIII-tag-CDKA;1 fusion construct. This is indeed what is mentioned in the referenced paper (Pusch 2012), but there's no mention of a "Strep III" tag in any other publication by that group or any others (or indeed, anywhere else on the internet). There is a well-established Strep II tag, which is perhaps what Pusch et al. meant? Please verify what exactly this tag is.

We have added the description of the StrepIII tag in the methods. StrepIII in Pusch et al. 2012 and here represents the Twin-strep-tag® developed by the IBA GmbH, which consists of two tandem Strep II tag moieties separated by a short linker. This tag shows better binding characteristics in comparison to Strep II tag (For more details, see publication below). The exact protein sequence of StrepIII/Twin-strep-tag is WSHPQFEK-GGGSGGGSGGSA-WSHPQFEK (underlines indicate the Strep II tag).

Schmidt T G M, Batz L, Bonet L, et al. Development of the Twin-Strep-tag® and its application for purification of recombinant proteins from cell culture supernatants[J]. Protein expression and purification, 2013, 92(1): 54-61.

There are several misspellings that would be caught by a spellchecker. For example:

Page 10, the word "restauration" should be "restoration"

Page 16, "essention" should be "essential"

We have corrected this.

Other typos:

Page 17, first line - "cohesion" should be "cohesin"

We corrected this.

Page 17, second paragraph - "evaded" should probably be "evicted"

We have corrected this.

Referee #2:

The manuscript entitled "The Arabidopsis Cdk1/Cdk2 homolog CDKA;1

controls chromosome axis assembly in meiosis", written by Yang et al., describes the role of CDKA;1 in phosphorylation and molecular dynamics of ASY1, the axial component of meiotic chromosomes in Arabidopsis. At first, to delimit the meiotic stage where CDKA;1 interacts with meiotic chromosomes, the authors introduced CDKA;1 variants with less kinase activity into *cdka;1* null mutants, and the hypomorphic mutant phenotypes and subcellular CDKA;1 localization suggested that CDKA;1 physically interacts with the chromosome axis during early meiotic prophase. Next, they examined the role of CDKA;1-dependent phosphorylation of ASY1 on meiotic chromosome-axis assembly, by introducing the fluorescent ASY1 fusion proteins with various mutations in ASY1 phosphorylation sites, and concluded that the phosphorylation site in the HORMA domain, T142, is major for phospho-regulation of ASY1, with the site T184 having an ancillary role.

In addition to the CDKA;1 role, the authors asked the role of ASY3 and PCH2 in ASY1 dynamics in axis assembly. From careful observations of localization patterns of ASY1 variants in the *pch2* mutant, they concluded that PCH2 plays dual roles and Y2H analyses; one is in helping ASY1 to accumulate into the nucleus, and another is in promoting the release of non-phospho ASY1 from the axis, and simultaneously, increasing the binding affinity of phospho ASY1 to ASY3. Furthermore, they proved the importance of the C-terminal closure motif in self-polymerization and chromosome association of ASY1.

All experiments were elaborately designed and performed, and the results were enough for supporting the conclusions. The information in this paper will be beneficial for all readers studying meiosis not only in plants but also in non-plant species. I wrote several minor comments below;

We are very thankful for the very positive feedback and the good comments of this reviewer.

(1) In Fig 1C pictures, it was difficult for me to confirm the localization of linear structure of CDKA;1 on DAPI-stained chromosomal threads. I think it

better to enlarge several regions and make co-localization clearer, or to show co-localization with some axial components, such as ASY1 and ASY3.

We agree with the reviewer that this is an important point that needs to be shown unambiguously. In this revised manuscript, we provided the co-localization of CDKA;1 with ASY1, also in response to one comment of reviewer 1, showing that CDKA;1 localizes to nonsynaptic chromosome regions.

(2) L20 and 21, p. 14: ASY 1-570  ASY1 1-570.

We corrected this.

(3) L25, p. 16: essention  essential?

We corrected this.

Referee #3:

The Arabidopsis Cdk1/Cdk2 homolog CDKA;1 controls chromosome axis assembly in meiosis

Chao Yang, Kostika Sofroni, Erik Wijnker, Yuki Hamamura, Lena Carstens, Hirofumi Harashima, Sara Christina Stolze, Daniel Vezon, Liudmila Chelysheva, Zsuzsanna Orban-Nemeth, Gaëtan Pochon, Hirofumi Nakagami, Peter Schlögelhofer, Mathilde Grelon and Arp Schnittger

It is now well established that the proteins that comprise the meiotic chromosome axis not only play an important role in the organization of the chromosomes during prophase I but are also central to the controlled formation of genetic crossovers. In many organisms including Arabidopsis, the central components of the chromosome axis have been identified

however many aspects of their functionality remain to be resolved. In previous studies the Schnittger group had demonstrated a role for the Cdk1/Cdk2 homolog, CDKA;1, during meiosis in Arabidopsis thaliana. In this manuscript Yang et al have further investigated the role of the kinase in relation to its interplay with the meiotic chromosome axis. As a result the work brings new insight into the regulation of the assembly of the axis. The work reveals that the HORMAD protein ASY1 is at least in part regulated by CDKA;1 phosphorylation and that this is important for the formation of the chromosome axis.

Specific comments

CDKA;1 is dynamically localized during meiosis:

i. The authors report a change in the comparative intensity of the CDKA;1:mVenus signal in the nucleus relative to the cytoplasm during prophase I. Based on the intensity levels it would seem that the maximum differential is around 1.5X. Some comment as to why and how the authors think this is of functional significance should be included.

At the moment, we can only speculate about the reasons and consequences of these changes in localization. In the discussion, we propose that high CDKA;1 activity in the nucleus early in meiosis is important/promotes phosphorylation of ASY1 (and possibly other processes needed for recombination). After this task is completed, CDKA;1 accumulates more in the cytoplasm, likely driven by another cyclin. What the substrates in the cytoplasm are, is unfortunately not clear yet. We also think that even small changes in CDKA;1 concentration are relevant to meiosis, please see for instance a recent paper in PNAS by Wijnker et al. (2019) in which two hypomorphic mutants in CDKA;1 showed substantially reduced recombination patterns with respect to each other.

ii. In Fig. 1c the authors use immunolocalization to show that CDKA;1 appears associated with the chromosome unsynapsed chromosome axis. Some regions appear to be devoid of signal and they state that these

correspond to the synapsed regions. They rely on the thickness of the DAPI staining to identify the synapsed regions. This may be the case, but to be certain they should use a synaptonemal complex specific marker, such as ZYP1 to confirm this point.

As explained above in response to reviewer one, we have technical problems to perform co-immunolocalization experiments with an anti-ZYP1 and an anti-Strep antibody. However, we hope that this reviewer agrees that the added co-immunostaining of CDKA;1 with ASY1 substantiates our observation that CDKA;1 is not present at synapsed regions (Figure 1C).

Meiosis is severely affected in hypomorphic *cdka;1* mutants:

i. The authors report that loss of CDKA;1 activity disrupts meiosis with a loss of crossover formation. Based on analysis of DAPI-stained chromosome spreads they report an absence of synapsis at the "zygotene-like stage". Inspection of panel 1Dh does however show regions of aligned chromosomes that have an intensity of staining not unlike the wild-type control, albeit much less complete. Moreover the authors then go on to say that at pachytene the chromosomes are "largely unpaired". This appears to contradict the previous statement. It does seem there is a synaptic defect however further characterization using a ZYP1 marker would clarify just how much, if any, synapsis is occurring. They should also stick to using the word synapsed rather interchangeably using synapsed or paired. The latter could refer to aligned chromosomes that are pre-synaptic or asynaptic.

Minor point: figure 1D requires a scale bar.

We thank the reviewer for helping in the clarifying the mutant description. We refer now, as suggested, to synapsed chromosomes throughout the manuscript. We also revisited the synapsis of the *CDKA;1^{T161D}* mutants by checking the localization of the synapsis marker ZYP1 using immunofluorescence. While full synapsis was observed in the male meiocytes of the wildtype at pachytene, as

seen by the complete co-alignment of ZYP1 with the chromosomes, no obvious staining of ZYP1 was observed in *CDKA;1^{T161D}* mutants, which suggests that the synapsis is largely (if not fully) abolished in *CDKA;1^{T161D}* mutants (Figure EV1E). Please note that *CDKA;1^{T161D}* mutants have very pleiotropic defects due to the many functions of this kinase in meiosis, please see also our discussion.

We added the scale bar for Figure 1D.

ii. The authors then go on to say that based on immunolocalization of MRE11 and DMC1 (again no scale bar) see Fig S1e, that these proteins are correctly localized suggesting DSBs are normal. However, they do not provide a wild-type control and there are no counts of foci. The figure indicates that there are foci on the chromatin but in the absence of an axis marker they cannot confirm that the foci are correctly localized. Importantly, they appear to have overlooked the previous observation by Lohmiller et al (Chromosoma 2008, 117; 277-288) that MRE11 forms foci in the absence of DSBs, hence it is not the best choice for demonstrating DSB formation. Ideally, this needs to be carried out using anti-gamma H2AX with proper counts.

The gamma H2AX antibody does not work very well on meiocytes in our hands and delivers a lot of background. We provide now the wild-type control and foci counts for the immunostaining of DMC1 showing that there is no significant difference between the wildtype and *CDKA;1^{T161D}* mutants (138.5 ± 9.8 n=10 vs 169.9 ± 15.7 n=7 in WT, p=0.09) (Figure EV1F). This suggests that the formation of DSBs is largely unaffected in *CDKA;1^{T161D}*.

Scale bars were added.

As suggested, we removed the immunolocalization of MRE11 that is not an ideal indicator for the DSB formation.

Similarly the double mutant with *rad51*, (no scale bars) does appear to show fragments but whether or not this is the same as the *rad51* single mutant cannot be ascertained in the absence of a control and counts. While I would agree that the data suggest DSBs are formed in the *cdka;1* mutant, the data provided is insufficient to conclude that breaks form as normal.

We now provide chromosome spread analyses of *rad51* single mutant as a control and sample sizes (Figure EV1G). A similar level of chromosome fragmentation between *rad51* single and *rad51 CDKA;1^{T161D}* double mutants was observed (44/45 in *rad51 CDKA;1^{T161D}* vs 39/39 in *rad51*).

Scale bars were added.

Phosphorylation of ASY1 by CDKA;1 is essential for its recruitment to the chromosome axis

i. The authors write that based on the mutant phenotype and the presence of a CDC28 phospho site in the yeast Hormad protein, Hop1 that this led them to identify ASY1 as a potential target for CDKA;1. They should also mention that a previous study of in vivo phosphorylation of ASY1 (Osman et al. Plant Journal 2016, 93, 17-33) had identified the protein as a potential CDK1 (and ATM/ATR) target. Although this work was based on the closely related species *Brassica oleracea*, the site in question, T535 is conserved in *Arabidopsis*. Of course, as it turns out, it does appear that based on this study T535 doesn't seem to be crucial for ASY1 function. That the other sites CDKA;1 sites identified here were not identified in the earlier study may simply be due incomplete coverage of the ASY1 in the MS analysis or perhaps indicate that the CDKA:1 is phosphorylation of ASY1 is relatively transient. Also it is worth pointing out the Osman study identified CDKA;1 as a putative ASY1 interacting protein.

We thank the reviewer for this comment and refer now to the work by Osman et al.: “Notably, a previous study identified the ASY1 orthologue of *Brassica oleracea*

as a potential in vivo ATM/ATR and CDK phosphorylation-target (Osman et al., 2017).”

ii. The authors commence by investigating the localization of a tagged ASY1 in the weak *cdka;1* mutant compared to wild-type. They report a delay in localization based on the extension of the axis associated ASY1 signal relative to the cell morphology and migration of the nucleolus to the side of the nucleus. Previous studies have determined the timing (in hours) of meiotic events based on labeling the meiocytes during S-phase with a marker such as BrdU or EdU rather than the approach used here. Can the authors be sure that the phenotypic events used to judge meiotic progression occur in the wild-type and *cdka;1* with the same timing? If so, do they have evidence to support this?

Staging of meiosis can indeed be difficult. However, cell morphology and migration of the nucleolus during meiosis have been previously used as the criteria to judge the progression/substages of meiosis in many studies (Wang *et al*, 2004; Yang *et al*, 2006; Stronghill *et al*, 2014; Nelms & Walbot, 2019; Prusicki *et al*, 2019), suggesting a higher reliability of these criteria.

Second, while it is not surprising that the duration of meiosis in *CDKA;1^{T161D}* and the other meiotic mutants, e.g., *tam*, is perhaps not exactly same as that in wildtype, the correlation of these basic meiotic processes, e.g., the change in cell shape, migration of the nucleolus, chromosome axis assembly et al., is not significantly altered in many meiotic mutants (Prusicki *et al*, 2019). Although some mutants, e.g., *ask1* and *swi1/dyad*, were shown to affect the migration of the nucleolus, the correlation of the rest criteria is not uncoupled giving a rather reliable indication of the meiotic stages (Wang *et al*, 2004; Yang *et al*, 2019). Furthermore, a possible migration defect of the nucleolus is apparently not present in *CDKA;1^{T161D}* (Figure 2A).

Importantly, we applied the same criteria used for evaluating ASY1 to investigate the localization of ASY3, another key component of the chromosome axis, in *CDKA;1^{T161D}* mutants. For ASY3, no obvious difference in comparison to the timing in wildtype was observed (Figure 2A), substantiating that the delayed

assembly of ASY1 observed in *CDKA;1^{T161D}* is not due to the uncoupling of the chromosome axis assembly with the phenotypic criteria used.

iii. The authors identify 5 putative CDKA;1 sites in ASY1 and provide evidence that two of these are phosphorylated in vitro by the kinase. Importantly, one of these T142, turns out to be the regulatory site. The other site corresponds to that identified in the in vivo analysis by Osman et al. Curiously, T184 which they later show to enhance the effect the of T142V mutation was not found to be phosphorylated by CDKA;1 in vitro. Do the authors have an explanation for this? CDKA;1

In this revised version, we further confirmed the functional relevance of the phosphorylation of T142 in ASY1 by constructing and analyzing new non-phosphorylatable versions of ASY1 (Figure EV4A) (please see also our response to reviewer 4).

As a part of this work, we also exchanged T184 with Ala and Gly residues. These experiments revealed that the position T184 is very sensitive to any exchanges to smaller amino acids, and interaction with ASY3 was largely reduced or nearly completely abolished with these new *asy1* mutant versions (Figure EV4A). This suggests that T184 is structurally very important. Unfortunately, we cannot currently estimate whether, or if so to what extent, this site can be phosphorylated *in vivo*.

iv. Functional analysis of mutant lines in which combinations of the phosphosites have been mutated provides good support for the key role of T142/T184. As well as impacts on fertility, the authors point to reduced ASY1 polymerization plus more nucleoplasmic signal (Fig 3B) and partial synapsis (Fig S6). With regard to figure 3B how was the reduction in linear ASY1 quantified? Ideally measurements are required particularly as the quality of images obtained by confocal microscopy lack clarity making it difficult for the reader. Also the use of ASY3 as a marker for synapsis is not ideal as it is not possible to fully follow synapsis with this marker

particularly the zygotene to pachytene transition. The use of either their SC marker ZYP1-GFP construct (Fig S3a) or an anti-ZYP1 Ab is far more definitive for staging this progress. The same comment applies to Fig S6, the use of DAPI-stained spreads to monitor the degree of synapsis is less than ideal.

To make the compromised chromosome localization of ASY1-T142V more visible, we now show signal distribution profiles (Figure 3C). These signal distribution profiles underline that the localization of ASY1-T142V is more diffuse in comparison to that of the wild-type version (more small peaks with low amplitudes in the mutants) (Figure 3C).

To further monitor the degree of synapsis, we performed immunostaining of ZYP1 in the ASY1 phospho-mutant plants. While only some foci-like signal were observed in ASY1-T142V T184V (*asy1*) mutants, short stretches of ZYP1 were detected in ASY1-T142V (*asy1*), indicating incomplete synapsis (Fig 3E). These results are also consistent with the chromosome spread data and the partial rescue of *asy1* by ASY1-T142V (Fig EV3 and Appendix Fig S3).

Phosphorylation of ASY1 increases its binding affinity with ASY3

i. The authors use Y2H to investigate the interaction of ASY1 and ASY3.

Their data indicate that the T142V mutation reduces the interaction and this is restored in the T142S and T142D versions. Does this imply that residue T142 is also phosphorylated in budding yeast? Given that budding yeast can phosphorylate some proteins from higher eukaryotes on T/S residues it does seem reasonable to infer that this is the case here.

However, to confirm this is the case the authors could provide evidence of the phosphorylation status of ASY1 in the yeast host.

The reviewer raises another good point here. We performed the Phos-tag SDS-PAGE of yeast extracts expressing the ASY1¹⁻³⁰⁰ and ASY1^{1-300/T142V} proteins followed by the western blotting analysis, which showed that ASY1¹⁻³⁰⁰ was phosphorylated, and the T142V mutation abolished this phosphorylation (Appendix Fig S2D).

Phosphorylation of ASY1 counteracts the action of PCH2 in early but not late prophase

i. The authors make the novel and interesting observation that the lack of chromosome localization of the ASY1-T142V/T184V construct observed in wild type and *asy1* mutant plants is not found in the absence of PCH2 and that it loads on to the chromosomes as efficiently as wild-type ASY1. They then go on to demonstrate using an ASY1-GFP construct that the tagged protein is found in both the nucleus and cytoplasm in the *pch2* mutant. They report a similar accumulation of ASY1-T142V/T184V in the *pch2* mutant. Based on this they suggest that PCH2 facilitates nuclear localization of ASY1. Given the observation that chromosomal localization of ASY1-T142V/T184V in the *pch2* mutant appears normal, does this suggest that PCH2 is not a major player in this respect of ASY1 nuclear localization or its role is non-essential? Or maybe its role is to facilitate ASY1 turnover in the cytoplasm rather than nuclear localization?

The chromosome localization of ASY1-T142V T184V *per se* in *pch2* mutants is indeed indistinguishable from the one of the wild-type version of ASY1 in the wildtype. However, the signal intensity of ASY1 and ASY1-T142V T184V in the nucleus of *pch2* mutants appears to be lower than the wild-type version of ASY1 in the wildtype when the same microscopy settings were used, likely due to an increased cytoplasmic retention (see Figure 5C). We therefore like to speculate that, in addition to counteracting the chromosome localization of ASY1 in early meiosis, PCH2 promotes/enhances the nuclear localization of ASY1 rather than regulates the turnover of ASY1.

Self-assembly of ASY1 through its C-terminal closure motif is affected by the phosphorylation in the HORMA domain.

This section deals with the control of axis assembly and the effect of phosphorylation on the ASY1 closure motif. Initially the authors define a region in the C-terminal of ASY1 that contains a potential closure motif as previously described in some other Horma proteins. However, they do not

mention the published work by the Corbett group (BioArchive 2018/Elife 2019) which has previously defined the ASY1 closure motif. This should be corrected. Nevertheless, they do go on to provide some interesting new insights into the role of the closure motif in relation to PCH2 activity. Furthermore they provide evidence to suggest that phosphorylation of the sites in the Horma region of ASY1 is important for interaction with the closure domain.

The paper from the Corbett lab was meanwhile published in eLife and we cite this work in our revised manuscript.

In summary the authors provide good evidence to indicate an important regulatory role for CDKA;1 in the regulation of ASY1 during axis assembly. As they conclude in their discussion it does seem that the role of the kinase is in the assembly of ASY1 in the axis. It appears not to be important for removal of ASY1 at the synaptic fork at the zygotene/pachytene transition. Whether it is important for maintenance of ASY1 in the axis prior to synapsis remains to be seen. They raise the possibility that the removal of ASY1 at zygotene may require other post-translational modifications. This notion is consistent with the studies by Osman et al who identified a number of other phosphosites within ASY1. This could be mentioned.

This is a very good point to discuss in light of our results and the findings from Osman et al., were included in our revised discussion.

As detailed above there are a few issues with the manuscript in its present form. Nevertheless, once these have been addressed, this is a valuable contribution to our understanding of how the formation of the meiotic chromosome axis is controlled in higher plants.

We very much appreciate the positive feedback and like to thank this reviewer

for his/her constructive comments.

Minor points

i. There are a number of typographical errors in the text.

We carefully checked the manuscript and corrected these errors.

ii. The absence of error bars on some figures needs to be corrected.

We added scale bars and error bars when applicable.

iii. Details of the ZYP1B-GFP construct are not provided

The details of ZYP1B-GFP reporter are shown in our recent publication (Yang et al. 2019). We cite this paper now.

Referee #4:

The manuscript by Yang et al uses genetic, cytological and biochemical approaches to analyze the role that the cyclin-dependent kinase, CDKA;1 plays in chromosome synapsis in Arabidopsis thaliana. They show that CKDA;1 localizes to meiotic nuclei at the same time as the HORMA domain containing axial element protein, ASY1. Furthermoe, hypomorphic alleles of CKDA;1 exhibit synapsis defects that are downstream of DSB and resection. Phenotypic analysis of the five putative Cdk sites in ASY1 revealed a role for T132 in the HORMA domain in chromosome synapsis, with a contributing effect for T184. Non-phosphorylatable alleles at these sites were generated by substituting valine for threonine and the double mutant showed defects in chromosomal localization in vivo, as well as

interaction with a second axial element protein, Red1 in genetic and biochemical assays. Negative charges at T132 and T184 resembled wild-type and even promoted ASY1-ASY3 binding in pulldown assays. Interestingly the chromosome localization defect of the asy1-T132V T184V mutant was rescued by mutation of the PCH2 gene. Finally, this paper shows that the HORMA domain of ASY interacts with a C-terminal fragment of ASY1, which the authors propose is a closure motif. These results are interesting and if the interpretations of the data are accurate would be a strong contribution to our understanding of meiotic chromosome behavior in plants. However, there are several concerns that need to be addressed. Most importantly, the authors need to show that it is the inability to be phosphorylated, and not just the valines, that are responsible for the defects (see below). In addition, the scholarship of the manuscript is poor with many citations missing, some that are irrelevant, and some that do not support the facts for which they are being cited.

We are grateful to the positive feedback of this reviewer and like to thank him/her for the constructive comments, too.

Major comments:

1. The conclusion that CDKA;1 is a "master regulator of meiosis" is too strong given that what is shown is that it regulates protein-protein interactions between ASY1 and itself and ASY3. While this is an important aspect of meiotic prophase, it is only one part of a highly regulated process. A master regulator should be at the top of the hierarchy that controls everything. For example, while Mer2 is constitutively phosphorylated by Cdk in yeast, its role is to provide the priming phosphorylation for the Cdc7-Dbf4 kinase that travels with the replication fork. In this situation, it is really DDK which is the regulator providing the readout for replication.

We corrected this statement and wrote as: CDKA;1 is an important regulator of meiosis.

2. Beginning of last paragraph on page 4: The authors are confusing the central element with the central region. The central element is a dark staining linear structure located midway between, and running parallel to, the lateral elements. The central element is comprised of different proteins from the central region. The central region is created by the insertion of perpendicular transverse filament proteins such as Zip1/ZYP1. It is the transverse filament proteins of the central region that hold the lateral elements together. Since many Zip1 dimers polymerize to make the central region, it would be more accurate to say the central region is formed by Zip1 oligomers.

The reviewer is absolutely right. We corrected this.

3. At the end of the discussion the authors state that they show that CDKA;1 is "especially important for chromosome pairing and synapsis". Pairing is functionally distinct from synapsis and is the early alignment of homologous chromosomes prior to synapsis. To determine whether pairing has occurred, FISH experiments are necessary to show that homologous sequences have aligned. These types of experiments were not done. I agree with the authors' conclusions that negative charges on ASY1 promote synapsis, but see no evidence to support the claim about pairing.

We have modified the sentence, and removed the statement that CDKA;1 is important for pairing.

4. Figure 1A and movie: The authors claim that CDKA;1 is "dynamically localized during meiosis" and that "in early prophase, CDKA;1mVenus is equally distributed between the nucleus and the cytoplasm". I don't see this and don't understand the diagrams trying to explain it. The cartoon for

early prophase in Figure 1A shows a beige rectangle with a green oval that contains within it a white oval. These different parts are not labeled in the figure legend. My assumption is the beige rectangle is the cytoplasm and the white circle is the nucleolus. It is clear in both the movie and Figure 1 A that there is more green color in the nucleus compared to either the cytoplasm or the nucleolus. I see some changes in abundance, but this is not the same as "dynamic localization" which would suggest the kinase is moving from one location to another.

We thank the reviewer for pointing out that we did not clearly label this figure and have corrected this (as the reviewer assumed).

We agree that we don't have single molecular resolution and cannot say that the same proteins move in and out of the nucleus. Hence, we refer to this pattern now as "changes in distribution".

5. In Figure 1C, the authors claim that CDKA;1 is removed from areas of the chromosomes that have synapsed. The basis for determining what is synapsed and what is not is not clear. This is an important conclusion since it correlates with the localization of ASY1 and is part of their argument that CDKA;1 regulates ASY1 localization. Therefore the relationship of CDKA;1 localization to chromosome synapsis should be determined by co-localization experiments with ZYP1 to show that the two do not co-localize (similar to the experiment that the authors did with ASY1 and ZYP1).

This point was also raised by two other reviewers, please see our detailed response there. In short, co-localization of CDKA;1 and ZYP1 is for technical reasons not possible in our hands. We have still tried to address this point by performing co-immunolocalization experiments with ASY1 and CDKA;1 and can clearly show that the regions where ASY1 is depleted (presumably due to synaptonemal complex formation) are also largely devoid of CDKA;1. We think that these experiments substantiate our initial claim that CDKA;1 is not present at synaptic regions.

6. Figure 1D shows a striking phenotype for the *cdka;1-D* mutant where there are 10 univalents instead of 5 bivalents (panels k vs d). After this statement it says (3%; n = 7). What does this mean? Did only 3% of cells exhibit this phenotype? What did the other 97% look like? Does n= 7 mean that 7 spreads were examined? If so, where does the 3% come from?

We thank this reviewer for pointing out this mistake. We have corrected it in this revised version. We have counted 9 metaphase I-like spreads with none of them showing any bivalents, suggesting a complete asynapsis of *CDKA;1^{T161D}* mutants.

7. It is not clear what is going on with the in vitro kinase assay in Figure 2C. According to the Figure legend, the top panel is an autoradiograph, suggesting that phosphorylation is being monitored by the transfer of a radioactive phosphate onto recombinant ASY1. However, in the methods section there is no mention of the use of radioactive ATP. Assuming I have interpreted the assay correctly, then there should be no signal for ASY1 in the autoradiograph in the absence of a CDK. However there is clear ASY1 band in Lane 1 and also in Lane 4 which the authors claim is a negative control. Further explanation is necessary for the reader to understand this experiment.

In this revised version, we labeled the figure clearly, and added a detailed description of this kinase assay using radioactive ATP in the method section of *In vitro* kinase assays. Indeed, there is background phosphorylation and we very clearly mention this. The reason for this is not entirely clear, most likely, kinase activity was co-purified from insect cells in this assay.

This kinase assay is a kind of a historical experiment, which we performed at the moment when we could not purify ASY1 from the *E. coli*. However, after quite a long time, we managed to obtain enough ASY1 proteins in good quality from *E.coli*, which we used thereafter for the remaining

experiments, e.g. in kinase assays with CDKA;1 and SDS to map the phospho-sites by mass spec.

8. Figure 4 legend: The title overinterprets the data and should be changed to something like "A negative charge at T132 in the HORMA domain of ASY1 promotes interaction with ASY3". The experiments in this panel use aspartic acid to mimic phosphorylation instead of assaying phosphorylation directly. Binding affinity is a specific parameter which needs to be measured biochemically. The quantification of the binding experiments in 4B is not clear. How was the ASY1 normalized? It should be to the amount of Red1 in the pulldown. 4C graphs "Relative levels of ASY1"-relative to what? What Triton-X 100 concentration was shown for the graph?

We have changed the title of the legend of Figure 4 to: A negative charge at T142 in the HORMA domain of ASY1 promotes its interaction with ASY3.

To measure the binding affinity biochemically, we sought to determine the dissociation constant (K_d) of the ASY1 variants with ASY3 using the Microscale thermophoresis (MST) analysis. While we found the K_d of ASY1¹⁻³⁰⁰-to-ASY3 (431 nM) and ASY1^{1-300/T142V T184V}-to-ASY3 (557 nM) interactions are similar, the K_d of ASY1^{1-300/T142D}-to-ASY3 (98 nM) interaction is much lower, suggesting a stronger binding of ASY1^{1-300/T142D} with ASY3 (see graph below). However, due to the large amount of protein (ligand) needed, we could only manage to perform the experiment once during this revision. As mentioned above, the purification of ASY1 and ASY3 is still challenging. Having no additional biological replicates, we show the results only below and not in the manuscript. Nonetheless, we hope this reviewer agree that the Y2H and *in vitro* GST pull-down experiments provide sufficient evidence to support our conclusion.

For the quantification of Figure 4B, we provide detailed information in the legend. The band intensity in the pull down of ASY1¹⁻³⁰⁰/T142V T184V at a Triton X-100 concentration of 0.5% was defined as 1. As suggested by this reviewer, the relative amount of ASY1 in the pull down fractions was normalized by the band intensity of the pulled-down ASY3. The average band intensity of ASY1 at different concentrations of Triton X-100 used was plotted.

9. An important question is whether the interpretation that defects observed for the ASY1 mutants with valine substitutions for CDK phosphosites is due to a lack of phosphorylation is correct. A recent paper (Winters and Pryciak, Mol Biol Cell 2019) showed that valine mutations in STE5 resulted in hyperactive signalling in the yeast pheromone response pathway but that this was not observed when the putative phosphosites were replaced with either alanine or glycine. The idea is that valine has a more hydrophobic side chain than serine or glycine and that this can affect protein function beyond not getting phosphorylated. This (hopefully unlikely) possibility can be tested by mutating T132 and T184 in the ASY1 2-hybrid construct and showing that this also abolishes the interaction with ASY3. If this is not the case, then the major conclusions of the paper are in doubt.

We thank the reviewer for making us aware of the effect of valine substitutions and the corresponding reference. For this revised version, we have exchanged

T142 and T184 to both alanine and glycine in single and double substitutions. These constructs were tested by yeast two-hybrid interaction assays in comparison to the valine mutation. Consistent with the valine mutation, we found that the mutations of both ASY1-T142A and T142G strongly reduced the interaction of ASY1 with ASY3, and that the mutation of ASY1-T142A T184A further decreased this interaction as that of ASY1-T142V T184V (Figure EV4A). Notably, while ASY1-T184V does not show any obvious effects alone with respect to the binding capacity of ASY1 to ASY3, ASY1-T184A largely reduced this interaction, and ASY1-T184G even abolished the binding to ASY3 (Figure EV4A). The reason for this reduction is not clear but it seems that T184 is a structurally important amino acid and this position does not tolerate a small amino acid. In any case, these additional substitutions are all consistent with our initial observation and support the conclusion that ASY1 needs to be phosphorylated at least at residue T142 to promote the interaction with ASY3.

10. Does the C-terminal ASY1 fragment proposed to contain the closure motif have sequence similarity with the closure motif consensus published by West et al? If so, that would make the authors' case more compelling.

As shown below, there is no high sequence similarity between the Hop1 and ASY1 closure motifs.

Hop1/585-605	-	-	-	-	F	D	E	S	V	P	A	K	I	R	K	I	S	V	S	K	K	T	L	K	S	N	W	
ASY1/571-593	D	R	R	G	R	K	T	S	M	V	R	E	P	I	L	Q	Y	S	K	R	Q	K	S	Q	A	N	-	-

During the submission and peer-review process, the paper by West et al. was published in eLife (before in bioRxiv) in which they describe the closure motif in the C-terminus of ASY1. In the revised version, we have cited and discussed their findings.

Minor comments:

Page 3, The Roeder lab paper describing Mer2 should be cited after the statement that it is a substrate for Cdk.

We added this.

References for the first sentence in paragraph 3 about Cdk phosphorylating Sae2 are needed.

We added this.

References are needed for the statement that "single stranded ends are captured by recombinases Dmc1 and Rad51....".

We included this and we wrote:.... single DNA strands are localized by the recombinases Rad51 and Dmc1 to promote strand invasion and formation of heteroduplex DNA (Shinohara *et al*, 1997; Kurzbauer *et al*, 2012; Da Ines *et al*, 2013).....

Line 31, Heteroduplex DNA is not "resolved", Holliday junctions are.

We corrected this and wrote as "Depending on how the subsequently resulting double Holliday junctions are resolved, meiotic crossovers....."

Page 4, top: The statement that "reducing Cdc28 activity...did not affect the number of Rad51 foci" contradicts the earlier assertion that Cdk activity is required to make DSBs. Also please indicate in which organisms these results were generated.

We have double-checked the relevant publications and the apparent contradiction is indeed due to our misinterpretation of a figure in Zhu et al., 2010. Therefore, we corrected the statement: Consistent with a partial co-localization with Rad51, inhibition of Cdk activity in early meiosis abolished the formation of RAD51 foci, leading to the conclusion that activity of Cdk is essential for DSB formation and/or processing (Henderson *et al*, 2006; Huertas *et al*, 2008; Zhu *et al*, 2010)

The authors should define Cdc28 as the catalytic subunit of Cdk in yeast. Then throughout, for example top of page 5, it would be more accurate to say phosphorylation by Cdk, not Cdc28, since Cdc28 is not active on its own.

We have followed this suggestion.

Line 13: The authors should cite the Rockmill and Roeder 1990 Genetics paper, and the Smith and Roeder 1997 JCB paper which showed that Red1 is part of the axial core and not on chromatin. Hollingsworth and Johnson is not relevant here.

We corrected this.

Line 15: The authors should cite the Hollingsworth et al 1990 Cell paper for Hop1 in yeast.

We cite this paper now.

Line 20: Smith and Roeder 1997 should be cited for the requirement of RED1 for Hop1 to bind to chromosomes. Again, Hollingsworth and Johnson is not relevant here.

We corrected this.

End of paragraph: Niu et al 2005 should be cited for the meiotic phenotypes of the hop1-K593A mutant.

We have added this paper.

The authors should cite the Toth 2009 PLoS Genetics paper for the loss of the HORMAD proteins from synapsed chromosomes which is dependent upon the Pch2 ortholog.

We have cited this paper.

Page 5, first paragraph and throughout: instead of "de-phospho" which is non-standard usage, say "unphosphorylatable".

We have changed the nomenclature throughout the manuscript.

Line 9; ...several Cdks...

We have corrected this.

A style suggestion to make the writing more concise-remove words like "has been found, ...were found...etc". So the second paragraph on page 5 could read: The model plant...and some of them function in meiosis. Six out of the ten...are expressed in meiosis.... However of these eight...and SDS exhibit meiotic defects."

We have largely adopted this style in the revised version.

Page 6, top: need refs for the relationship of CDKG to human Cdk10 and for reason why it is not part of the core cell cycle machinery.

What we meant was that CDKG likely does not phosphorylate proteins involved in chromosome organization. The transcriptional targets could very well be core cell cycle regulators. We have tried to make this point clear now.

Page 7, line 6, "After synapsis of the homologs".

We have corrected this.

Last paragraph, please indicate some of the mitochondria shown in Figure 1D with arrows.

We have added arrows.

Page 8: first paragraph, "The absence of synapsis and...."

We have corrected this.

Line 26, saying that CDKA;1 is "indispensable...for chromosome synapsis and bivalent formation" is too strong given that it appears that many cells in the mutant are capable of these processes

We rephrased the sentence and wrote as: Taken together, these data suggest that

CDKA;1 is an important regulator of meiosis especially for chromosome synapsis and bivalent formation.

Line 30, again, pairing was not assayed and therefore cannot be claimed to be affected.

We have removed this statement.

Page 9, line 5: needs a citation for the fact that *asy1* mutants are asynaptic

We have added this.

Line 11, In yeast, Hop1 and Red1 were shown to localize to the axis, not to chromatin (see the Smith and Roeder 1997 paper). The chromatin is in the loops that emanate out of from the axis. Do the authors think that ASY1 binds to chromatin as well as the axis?

We rephrased the sentence and wrote that ASY1 localizes to the chromosome axis.

Page 10, top "Successful" not "Succeeding"

We have corrected this.

Line 3, remove "strongly" phosphorylated since no quantification is presented. Also see Major comment 7 about my confusion with the interpretation of this experiment.

We have removed this.

Line 19: ...was unaffected in the plants expressing...

We have changed this.

Line 20: not clear what is meant by the verb "collate" in this context

We rephrased and wrote as: Consistent with its chromosomal loading being independent on *ASY1*, the expression and localization of *ASY3* was unaffected in the plants producing different *ASY1* variants and hence, was used in the following as a marker for the staging of meiosis (Figure 3B).

Line 24: restoration, not restauration

We have corrected this.

Line 25: silique not siliques

We have corrected this.

Line 27: ...during leptotene similarly to *ASY:GFP*...

We have changed this.

Line 32: Cite figure for failure of the 5V mutant to localize

We cite this now.

Page 11, line 4, wild-type

We have corrected this.

Line 21, ...was no longer phosphorylated...

We have corrected this.

Page 12, line 2: delete "strongly"

We have deleted it.

Line 7, it would be helpful if the authors explained the details of their two hybrid assay. It appears there are two reporters and that the single -His selection is less stringent than the double selection for histidine and adenine protorophy, which is why they conclude that the T142V T184V mutant has a more severe interaction defect than T142 alone.

We have added further information to explain our assay in the main text.

Figure 4A: It is interesting that the T142 and S142 ASY1 truncations interact with ASY3 in the two-hybrid system, since this suggests that these amino acids are phosphorylated in mitotic yeast cells (unless it is because they are not valines). Therefore the specificity of the cyclin does not appear to be critical for this modification.

In this revised version, we show that the mutations of T142 to both alanine and glycine largely reduce its interaction with ASY3, which suggests that the

decrease of interaction is unlikely attributed to the valine, but to the inability of phosphorylation. We further confirmed that T142 was phosphorylated in yeast cells (Appendix Fig S2D). We agree that this suggests that the specificity of the cyclin might be not critical for this modification, at least in yeast. Possibly, the cyclin partner, e.g., SDS, promotes the localization of CDKA;1 to its site of action *in vivo*, e.g., to the chromosome axis. Clearly, further work is required to explore this aspect but we hope that the reviewer agrees that this exploration goes beyond the scope of this work.

Line 26, phospho-mimicking, not phospho-mimicry

We have corrected this.

Page 13, line 13, necessity

We have corrected this.

Page 14, top, Need REFs for chromosomal localization of Hop1 and the HORMADs in yeast (Smith and Roeder, 1997 and Toth paper).

We have cited these papers now.

Bottom, To my eye, the growth of the T142D mutant is not as robust as the wild-type, so saying the two showed "similar interaction strength" is not consistent with the data.

We altered the sentence and wrote as: Conversely, the phosphorylation-mimicking version ASY1^{1-300/T142D} showed higher interaction strength despite a slight decrease compared to that of the non-mutated ASY1 version.

Page 16, line 27, essential, not essention

We have corrected this.

Page 17, line 4: the authors say: "...binding capacity...was enhanced when the closure motif of ASY1 was deleted". This is an over-interpretation of the data. Binding capacity was enhanced when a terminal region of ASY1 was deleted. The authors hypothesize that this is because the terminal domain contains a closure motif but this has not been proven. If this region contains the sequence determinants previously shown to be present in a closure motif, then this conclusion would be warranted.

We rephrased the sentence and wrote as: The binding capacity of ASY1 to ASY3 was strongly enhanced when a short C-terminal region including the presumptive closure motif of ASY1 was deleted (Figure 6B).

Line 9, What does "at least in yeast" mean in this sentence which is talking about ASY1 interaction?

We clarified this and wrote as: at least when being expressed in yeast.

Second paragraph, It is not true that in yeast PCH2 is required for synapsis and recombination. Several papers have shown that pch2 mutants in yeast exhibit more crossovers than wild-type (one of them being the Joshi et al reference that is cited). Synapsis also occurs, but instead of alternating Hop1 and Zip1 domains, Zip1 staining is continuous.

We have corrected this.

Line 20, not sure what "evaded" means in the context of this sentence

This was a typo, we corrected this to "evicted".

Page 18, line 2 not sure what "partition" means in the context of the sentence

We rephrased and wrote as: However, the *sds* mutant phenotype is not a subset of the phenotype of the weak loss-of-function *cdka;1* mutants as seen by the apparently correct localization of DMC1 in *CDKA;1^{T161D}* and the localization failure in *sds* (DeMuyt et al., 2009).

Bibliography, gene and genus species names should be italicized. Only proper nouns and the first word of the title should be capitalized.

We have checked this throughout the manuscript and corrected when necessary.

Thank you for submitting your revised manuscript for our consideration. Three of the original referees have now looked at it again, and I am pleased to say that they consider the key scientific issues well addressed and the study now in principle suitable for publication. However, referee 4 notes that several previously raised concerns have not been well-addressed throughout the text, therefore requiring further modifications. In addition, referee 1 also asks for some additional text/discussion changes. I would therefore like to invite you to incorporate those changes - using the text processors "track changes" function - during a final round of minor revision.

In addition, there a number of editorial issues should also be addressed at this stage.

REFRE REPORTS

Referee #1:

The authors have addressed my concerns adequately, and to my eye also admirably addressed the large number of comments from the other reviewers. One change I would suggest is in the abstract: I'm not sure the authors can claim to have identified the closure motif in ASY1 if this has already been reported; I'd suggest rephrasing to focus on the authors new observations that ASY1 phosphorylation appears to affect the HORMA-closure motif interaction.

With respect to reviewer 4's question about whether the ASY1 closure motif contains the closure motif consensus identified by West et al, it's important to note that closure motifs bound by different kingdoms' meiotic HORMADs (e.g. fungi vs. plants vs. animals) seem to show no identifiable sequence similarity. Thus while it is valid to ask whether each species' Red1 and Hop1 equivalents contain similar closure motifs (which would bind the same protein), comparisons between, say, yeast and plants are not very informative.

I am fine with the authors sharing my comments about the likely structural location of T142 in the ASY1 HORMA domain structure. I just also looked at the PHYRE prediction again to see whether T184 is predicted to be in an interesting place. It seems like this residue is likely to be located early in the protein's "safety belt", and the structure prediction does not provide a clear prediction for why its mutation to a smaller residue would not be tolerated.

Referee #3:

The Arabidopsis Cdk1/Cdk2 homolog CDKA;1 controls chromosome axis assembly in meiosis
Chao Yang, Kostika Sofroni, Erik Wijnker, Yuki Hamamura, Lena Carstens, Hirofumi Harashima, Sara Christina Stolze, Daniel Vezon, Liudmila Chelysheva, Zsuzsanna Orban-Nemeth, Gaëtan Pochon, Hirofumi Nakagami, Peter Schlögelhofer, Mathilde Grelon and Arp Schnittger

This revised manuscript provides a valuable contribution to understanding the role of CDKA;1 during meiotic prophase I progression in Arabidopsis thaliana. The authors have addressed the concerns I raised in relation to the previous version of their manuscript. In all but one case these concerns have been dealt with in full. Unfortunately, the authors have not been able to perform dual-immunolocalization of ZYP1 and CDKA;1 that would confirm that the latter is absent on synapsed chromosomes. This was a point also raised by another reviewer. Nevertheless, they do show that a reduction of CDKA;1 signal is coincident with depletion of ASY1 on chromosome regions that appear synapsed. Since it is well established in the literature that ASY1 is depleted from the chromosome axes as they synapse during zygotene, I think it is reasonable to accept that their conclusion regarding the CDKA;1 signal is in all likelihood correct despite the somewhat indirect route. Overall, this a substantial piece of work.

Referee #4:

The manuscript by Yang et al uses genetic, cytological and biochemical approaches to analyze the roles that the cyclin-dependent kinase, CDKA;1 plays in chromosome synapsis in *Arabidopsis thaliana*. They show that CDKA;1 localizes to meiotic nuclei at the same time as the HORMA domain containing axial element protein, ASY1. Furthermore, hypomorphic alleles of CDKA;1 exhibit synapsis defects that are downstream of DSB and resection. Phenotypic analysis of the five putative Cdk sites in ASY1 revealed a role for T132 in the HORMA domain in chromosome synapsis, with a contributing effect for T184. Non-phosphorylatable alleles at these sites were generated by substituting valine for threonine and the double mutant showed defects in chromosomal localization *in vivo*, as well as interaction with a second axial element protein, Red1 in genetic and biochemical assays. Negative charges at T132 and T184 resembled wild-type and even promoted ASY1-ASY3 binding in pulldown assays. Interestingly the chromosome localization defect of the asy1-T132V T184V mutant was rescued by mutation of the PCH2 gene. Finally, this paper shows that the HORMA domain of ASY1 interacts with a C-terminal fragment of ASY1, which the authors propose is a closure motif. These results are interesting and make a strong contribution to our understanding of meiotic chromosome behavior in plants.

Given that the authors indicated in their response to reviewers that the comments/corrections from my previous review had been addressed, I was very disappointed to see when reading the manuscript many instances where this was not true. Prior to being accepted for publication, the authors should correct the numerous errors that this manuscript still contains.

1. My previous review noted that the authors did not understand the difference between the central element of the SC. Despite agreeing that I was "absolutely right" about this, at the top of page 5 the authors say that "homologs become connected in the SC via central elements formed by dimers of the Zip1/ZYP1 family...". This is a misconception that must be corrected.
2. My previous review noted that in the Bibliography, gene and genus species names should be italicized and that only proper nouns should be capitalized. The authors indicated this was "corrected when necessary" but numerous references in the Bibliography still contain one or more of these mistakes.
3. On page 4, please include a reference for the HORMA domain-Aravind and Koonin in Trends Biochem. Sci 1998.
4. Both I and another reviewer indicated that the Niu et al 2005 MBoC paper needed to be cited for the characterization of the hop1-K593A mutant. Instead the authors have referenced an unrelated paper from 2015 with a different author named Niu. The correct citation needs to be used.
5. In the last paragraph on page 4 the authors claim that "Hop1/ASY1 is recruited to the axis by direct interaction with Red1" and cite papers based on cytological analyses. These types of experiments do not prove that the interaction is direct, nor that the proteins interact, only that they co-localize. I recommend changing this to say that Hop1..is recruited to the axis by interaction with Red1..." and cite de los Santos and Hollingsworth 1998 and Bailis and Roeder 1998-these are the papers that first showed that Hop1 and Red1 co-immunoprecipitate from meiotic cells.
6. Page 5, second paragraph: As I pointed out in my previous review, Cdc28 is not an ortholog of Cdk. Cdk stands for "cyclin-dependent kinase" and is comprised of two subunits: a catalytic subunit and a regulatory subunit (the cyclin). Cdc28 is an ortholog of the catalytic subunit of Cdk.
7. The authors state that Figure 1A and B show that CDKA.1 is "equally distributed" between the nucleus and cytoplasm. This is clearly not the case or one wouldn't be able to make out the nucleus based on CDKA.1 staining (as one can). Furthermore the quantification in 1B shows that 60% of the CDKA.1 is in the nucleus in early prophase and the error bars indicate this is significantly different from 50%. This error was repeated later in the discussion and should be corrected there as well.
8. page 7, middle: Since the Ashley and Zhu references did not use plants, say "...show a distinct punctate staining in meiosis in mice and yeast.."

9. I found the language in this section confusing and suggest the authors be more explicit. "At zygotene, when homologous chromosomes start to synapse, the ASY1 and CDKA.1 signals coordinately disappeared. Since Asy1 is only present on unsynapsed chromosomes, we conclude from the similar staining patterns of Asy1 and CDKA.1 that the kinase is excluded from synapsed regions as well."

10. page 8 top: Zygotene is defined by the presence of partially synapsed chromosomes. It is therefore confusing that the authors state "...becomes notable at zygotene-like stage when no homolog synapsis is observed". This point needs to be clarified. Are the authors using some parameter other than partial synapsis to define a "zygotene-like stage"-if so what is it?

11. The authors have done a better job of explaining the kinase assay using the insect cell derived ASY1. However, it is still not explicit how they distinguished the CDK-dependent phosphorylation from the background phosphorylation. Are the authors suggesting there is more radioactivity incorporated when a Cdk1 is included in the reaction? If so, these numbers need to be quantified. Given that the experiment using the bacterially produced protein is more conclusive, I suggest removing the "historical" experiment which is not convincing.

12. The authors checked to see whether mutation of the putative Cdk phosphosites in ASY1 to amino acids other than valine reduced the interaction ASY3. I am glad to see that this was the case. However, to come to this conclusion, the authors need to show that these mutant proteins are as stable as the wild-type ASY1 protein, as was done for the other mutants.

13. page 14, 7 lines from the bottom: the word "that" is repeated

14. page 17, second line: "Cdk mutants" doesn't make sense. First, it is genes that are mutated and analyzed, not proteins. Second, the mutations must be in a gene that encode either the catalytic subunit or the cyclin-which was it?

15. Second paragraph: ...chromosome synapsis and bivalent formation are... DMC1 (*italics*)-dependent...

16. In my previous review, I noted that in yeast Hop1 and Red1 are localized to the axis and not the chromatin loops and asked whether the authors think that Asy1 is differently localized to chromatin (since this is what they said in the original manuscript). Their response was to say that the manuscript was corrected to state that "ASY1 localized to the chromosome axis". Yet at the bottom of page 17-18, the authors state that CDKA.1 acts directly at the chromatin...." and "has a more continuous appearance along chromatin resembling the localization of ASY3 and ASY1 itself." It is clear that the authors do not yet understand that the axis is the proteaceous base upon which the chromatin loops are tethered.

17. middle of page 18: say "unsynapsed" and not "non-synaptic"

Detailed response to reviewer and editorial comments on Yang et al.

Modifications for the figures:

Figure 1D: corrected the miss labeling of *D* to *CDKA;1^{T161D}*.

Figure 2B: added the quantification of band intensity.

Figure 4B: deleted the unnecessary red-dotted lines.

Figure EV1C: added a scale bar.

Appendix Figure S2E: added the data requested by referee 4 (comment 12).

Source data for Appendix Figure S2E: added none-cropped images.

Referee #1:

The authors have addressed my concerns adequately, and to my eye also admirably addressed the large number of comments from the other reviewers. One change I would suggest is in the abstract: I'm not sure the authors can claim to have identified the closure motif in ASY1 if this has already been reported; I'd suggest rephrasing to focus on the authors new observations that ASY1 phosphorylation appears to affect the HORMA-closure motif interaction.

We are not entirely sure how to address this point and would like to keep this decision to the editor. The paper by West et al. from the Corbett lab was deposited in BioRxiv shortly before we submitted our work. During the review process this paper was apparently accepted and subsequently published in eLife. This time frame shows that we have independently and genuinely identified the closure motif in ASY1. This comment also did not come up in the previous revision round. However, we are also okay with removing the corresponding sentence in the abstract.

With respect to reviewer 4's question about whether the ASY1 closure motif contains the closure motif consensus identified by West et al, it's important to note that closure motifs bound by different kingdoms' meiotic HORMADs (e.g. fungi vs. plants vs. animals) seem to show no identifiable sequence similarity. Thus while it is valid to ask whether each species' Red1 and Hop1 equivalents contain similar closure motifs (which would bind the same protein), comparisons between, say, yeast and plants are not very informative.

We agree with the reviewer. The protein sequence comparison of the closure motifs in the yeast HOP1 and *Arabidopsis* ASY1 shown in the previous “response to the reviewer comments” was just used for answering the question from reviewer 4 and is not included in this manuscript.

I am fine with the authors sharing my comments about the likely structural location of T142 in the ASY1 HORMA domain structure. I just also looked at the PHYRE prediction again to see whether T184 is predicted to be in an interesting place. It seems like this residue is likely to be located early in the protein's "safety belt", and the structure prediction does not provide a clear prediction for why its mutation to a smaller residue would not be tolerated.

We are very grateful to this reviewer for this positive feedback and the kind sharing of ideas, and would like thank his/her for his/her contribution in the review process.

Referee #3:

The Arabidopsis Cdk1/Cdk2 homolog CDKA;1 controls chromosome axis assembly in meiosis

Chao Yang, Kostika Sofroni, Erik Wijnker, Yuki Hamamura, Lena Carstens, Hirofumi Harashima, Sara Christina Stolze, Daniel Vezon, Liudmila Chelysheva, Zsuzsanna Orban-Nemeth, Gaëtan Pochon, Hirofumi Nakagami, Peter Schlögelhofer, Mathilde Grelon and Arp Schnittger

This revised manuscript provides a valuable contribution to understanding the role of CDKA;1 during meiotic prophase I progression in *Arabidopsis thaliana*. The authors have addressed the concerns I raised in relation to the previous version of their manuscript. In all but one case these concerns have been dealt with in full. Unfortunately, the authors have not been able to perform dual-immunolocalization of ZYP1 and CDKA;1 that would confirm that the latter is absent on synapsed chromosomes. This was a point also raised by another reviewer. Nevertheless, they do show that a reduction of CDKA;1 signal is coincident with depletion of ASY1 on chromosome regions that appear synapsed. Since it is well established in the literature that ASY1 is depleted from the chromosome axes as they synapse during zygotene, I think it is reasonable to accept that their conclusion regarding the CDKA;1 signal is in all likelihood correct despite the somewhat indirect route. Overall, this a substantial piece of work.

We appreciated very much the positive feedback from this reviewer and would like to thank his/her constructive comments on our paper.

Referee #4:

The manuscript by Yang et al uses genetic, cytological and biochemical approaches to analyze the roles that the cyclin-dependent kinase, CDKA;1 plays in chromosome synapsis in Arabidopsis thaliana. They show that CDKA;1 localizes to meiotic nuclei at the same time as the HORMA domain containing axial element protein, ASY1. Furthermore, hypomorphic alleles of CDKA;1 exhibit synapsis defects that are downstream of DSB and resection. Phenotypic analysis of the five putative Cdk sites in ASY1 revealed a role for T132 in the HORMA domain in chromosome synapsis, with a contributing effect for T184. Non-phosphorylatable alleles at these sites were generated by substituting valine for threonine and the double mutant showed defects in chromosomal localization in vivo, as well as interaction with a second axial element protein, Red1 in genetic and biochemical assays. Negative charges at T132 and T184 resembled wild-type and even promoted ASY1-ASY3 binding in pulldown assays. Interestingly the chromosome localization defect of the asy1-T132V T184V mutant was rescued by mutation of the PCH2 gene. Finally, this paper shows that the HORMA domain of ASY interacts with a C-terminal fragment of ASY1, which the authors propose is a closure motif. These results are interesting and make a strong contribution to our understanding of meiotic chromosome behavior in plants.

Given that the authors indicated in their response to reviewers that the comments/corrections from my previous review had been addressed, I was very disappointed to see when reading the manuscript many instances where this was not true. Prior to being accepted for publication, the authors should correct the numerous errors that this manuscript still contains.

We thank the reviewer for his continued constructive comments and apologize that some of the previous concerns have not be adequately integrated in the manuscript.

1. My previous review noted that the authors did not understand the difference between the central element of the SC. Despite agreeing that I was "absolutely right" about this, at the top of page 5 the authors say that "homologs become connected in the SC via central elements formed by

dimers of the Zip1/ZYP1 family...". This is a misconception that must be corrected.

We only corrected this in one part of the manuscript. We have now checked carefully the entire paper and have corrected this at all instance.

2. My previous review noted that in the Bibliography, gene and genus species names should be italicized and that only proper nouns should be capitalized. The authors indicated this was "corrected when necessary" but numerous references in the Bibliography still contain one or more of these mistakes.

We apologize that we have forgotten to double-check the references after formatting the manuscript. The style errors were in our bibliography program. In this revised version, we have carefully checked all the references.

3. On page 4, please include a reference for the HORMA domain-Aravind and Koonin in Trends Biochem. Sci 1998.

We included this paper.

4. Both I and another reviewer indicated that the Niu et al 2005 MBoC paper needed to be cited for the characterization of the hop1-K593A mutant. Instead the authors have referenced an unrelated paper from 2015 with a different author named Niu. The correct citation needs to be used.

We have corrected it.

5. In the last paragraph on page 4 the authors claim that "Hop1/ASY1 is recruited to the axis by direct interaction with Red1" and cite papers based on cytological analyses. These types of experiments do not prove that the interaction is direct, nor that the proteins interact, only that they co-localize. I recommend changing this to say that Hop1..is recruited to the axis by interaction with Red1..." and cite de los Santos and Hollingsworth 1998 and Bailis and Roeder 1998-these are the papers that first showed that Hop1 and Red1 co-immunoprecipitate from meiotic cells.

We have followed the suggestion of this reviewer and cite the papers of De los Santos and Hollingsworth 1999 and Bailis and Roeder 1998 in the revised version.

6. Page 5, second paragraph: As I pointed out in my previous review, Cdc28 is not an ortholog of Cdk. Cdk stands for "cyclin-dependent kinase" and is comprised of two subunits: a catalytic subunit and a regulatory subunit (the cyclin). Cdc28 is an ortholog of the catalytic subunit of Cdk.

We have made a more explicit description wrote as: ... Cdk complexes have also been implicated in the assembly of the SC since mutations in the catalytic core, i.e., in *Cdk2* in mice and in *Cdc28* (*Cdk1* homolog) in budding yeast, resulted in defects in SC formation (Ortega *et al*, 2003; Zhu *et al*, 2010)....

7. The authors state that Figure 1A and B show that CDKA.1 is "equally distributed" between the nucleus and cytoplasm. This is clearly not the case or one wouldn't be able to make out the nucleus based on CDKA.1 staining (as one can). Furthermore the quantification in 1B shoes that 60% of the CDKA.1 is in the nucleus in early prophase and the error bars indicate this is significantly different from 50%. This error was repeated later in the discussion and should be corrected there as well.

We have corrected this.

8. page 7, middle: Since the Ashley and Zhu references did not use plants, say "...show a distinct punctate staining in meiosis in mice and yeast.."

We have followed this suggestion in this revised version.

9. I found the language in this section confusing and suggest the authors be more explicit. "At zygotene, when homologous chromosomes start to synapse, the ASY1 and CDKA.1 signals coordinately disappeared. Since Asy1 is only present on unsynapsed chromosomes, we conclude from the similar staining patterns of Asy1 and CDKA.1 that the kinase is excluded from synapsed regions as well."

We followed this suggestion.

10. page 8 top: Zygotene is defined by the presence of partially synapsed chromosomes. It is therefore confusing that the authors state "...becomes notable at zygotene-like stage when no homolog synapsis is observed". This point needs to be clarified. Are the authors using some parameter other than partial synapsis to define a "zygotene-like stage"-if so what is it?

We have described the criteria that were used to judge the zygotene-like stage when synapsis is abolished. We write now: ...In *CDKA;I^{T161D}*, the first difference from the wildtype becomes notable at zygotene-like stage manifested by the presence of clear thread-like chromosomes and the accumulation of mitochondria in one side of the meiocytes, in which no homolog synapsis is observed (Fig 1D h) (58%; n=120)....

11. The authors have done a better job of explaining the kinase assay using the insect cell derived ASY1. However, it is still not explicit how they distinguished the CDK-dependent phosphorylation from the background phosphorylation. Are the authors suggesting there is more radioactivity incorporated when a Cdk1 is included in the reaction? If so, these numbers need to be quantified. Given that the experiment using the bacterially produced protein is more conclusive, I suggest removing the "historical" experiment which is not convincing.

While we agree with this reviewer that this is not a perfect experiment due to the background phosphorylation using ASY1 purified from insect cells, the results do show that there is a significant increase in phosphorylation of ASY1 upon the addition of CDKA;1-SDS and CDKA;1-TAM kinase complexes. This experiment furthermore suggests that ASY1 can be phosphorylated by different but not all CDKA;1-cyclin complexes. We have added the quantification for the band intensity in this revised version.

12. The authors checked to see whether mutation of the putative Cdk phosphosites in ASY1 to amino acids other than valine reduced the interaction ASY3. I am glad to see that this was the case. However, to come to this conclusion, the authors need to show that these mutant proteins are as stable as the wild-type ASY1 protein, as was done for the other mutants.

In this revised version, we have shown that the mutant versions of ASY1 (ASY1^{1-300/T142A}, ASY1^{1-300/T184A}, ASY1^{1-300/T142A;T184A}, ASY1^{1-300/T142G}, ASY1^{1-300/T184G}, and ASY1^{1-300/T142G;T184G}) were as stable in yeast cells as the wild-type version (Appendix Fig S2E).

13. page 14, 7 lines from the bottom: the word "that" is repeated

We have corrected this.

14. page 17, second line: "Cdk mutants" doesn't make sense. First, it is genes that are mutated and analyzed, not proteins. Second, the mutations must be in a gene that encode either the catalytic subunit or the cyclin- which was it?

In this context, we are talking the mutants of *Cdk1/Cdk2* in mice (meaning the names of the kinase genes in mice).

15. Second paragraph: ...chromosome synapsis and bivalent formation are... DMC1 (*italics*)-dependent...

We have corrected this.

16. In my previous review, I noted that in yeast Hop1 and Red1 are localized to the axis and not the chromatin loops and asked whether the authors think that Asy1 is differently localized to chromatin (since this is what they said in the original manuscript). Their response was to say that the manuscript was corrected to state that "ASY1 localized to the chromosome axis". Yet at the bottom of page 17-18, the authors state that CDKA.1 acts directly at the chromatin...." and "has a more continuous appearance along chromatin resembling the localization of ASY3 and ASY1 itself." It is clear that the authors do not yet understand that the axis is the proteineaceous base upon which the chromatin loops are tethered.

We have corrected this in this revised version.

17. middle of page 18: say "unsynapsed" and not "non-synaptic"

We corrected this.

Accepted

4th September 2019

Thank you for submitting your final revised manuscript for our consideration. I am pleased to inform you that we have now accepted it for publication in The EMBO Journal.

YOU MUST COMPLETE ALL CELLS WITH A PINK BACKGROUND ↓
PLEASE NOTE THAT THIS CHECKLIST WILL BE PUBLISHED ALONGSIDE YOUR PAPER

Corresponding Author Name: Arp Schnittger
Journal Submitted to: THE EMBO JOURNAL
Manuscript Number: EMBOJ-2019-101625